# Fine interaction profiling of VemP and mechanisms responsible for its translocation-coupled arrest-cancelation

**Ryoji Miyazaki, Yoshinori Akiyama, Hiroyuki Mori***

Institute for Frontier Life and Medical Sciences, Kyoto University, Kyoto, Japan

**Abstract** Bacterial cells utilize monitoring substrates, which undergo force-sensitive translation elongation arrest, to feedback-regulate a Sec-related gene. *Vibrio alginolyticus* VemP controls the expression of SecD/F that stimulates a late step of translocation by undergoing export-regulated elongation arrest. Here, we attempted at delineating the pathway of the VemP nascent-chain interaction with Sec-related factors, and identified the signal recognition particle (SRP) and PpiD (a membrane-anchored periplasmic chaperone) in addition to other translocon components and a ribosomal protein as interacting partners. Our results showed that SRP is required for the membrane-targeting of VemP, whereas PpiD acts cooperatively with SecD/F in the translocation and arrest-cancelation of VemP. We also identified the conserved Arg-85 residue of VemP as a crucial element that confers PpiD-dependence to VemP and plays an essential role in the regulated arrest-cancelation. We propose a scheme of the arrest-cancelation processes of VemP, which likely monitors late steps in the protein translocation pathway.

*For correspondence:
hiromori@infront.kyoto-u.ac.jp

**Competing interests:** The authors declare that no competing interests exist.

## Introduction

In bacteria, the evolutionarily conserved Sec translocon and SecA play essential roles in protein translocation across and integration into the membrane (*Rapoport et al., 2017*). The translocon consists of three membrane-integrated proteins, SecY, SecE, and SecG (*Mori and Ito, 2001*). SecY forms a channel-like path for substrate secretory proteins (*Van den Berg et al., 2004*) and is stabilized by SecE (*Taura et al., 1993*). SecG peripherally binds to SecY (*Tanaka et al., 2015*) and stimulates the SecA-dependent protein translocation (*Nishiyama et al., 1994*). SecA, an essential ATPase (*Lill et al., 1989*), binds to cytoplasmic protrusions in SecY (*Mori and Ito, 2006a*; *Zimmer et al., 2008*) and pushes a polypeptide into the translocon (*Economou and Wickner, 1994*; *Erlandson et al., 2008*). Secretory proteins are generally synthesized as a precursor with an N-terminal signal sequence recognized by the translocation machinery, including the translocon and SecA, and is cleaved off during translocation. Although bacterial protein secretion to the periplasmic space is thought to occur post-translationally in many cases, some secretory proteins use the signal recognition particle (SRP) pathway, which targets the nascent precursor protein to the translocon (*Huber et al., 2005*; *Shimohata et al., 2005*). Notably, even in the latter cases, translocation requires the SecA motor function (*Shimohata et al., 2005*). The Sec translocon is also utilized for protein integration into the membrane. Membrane proteins typically have no cleavable signal sequence, but their transmembrane (TM) regions act as a signal for integration. SRP binds to the signal sequence equivalent (a TM segment) of a membrane protein and deliverers a ribosome-nascent protein complex (RNC) to the translocon (*Steinberg et al., 2018*). The RNC binds to the same SecY region as SecA (*Frauenfeld et al., 2011*; *Kuhn et al., 2011*). Bacteria also have the SecD/F complex that stimulates protein secretion in vivo (*Pogliano and Beckwith, 1994*) and facilitates the release of a polypeptide from the Sec translocon during a late step in translocation (*Matsuyama et al., 1993*). Structural and functional analyses have shown that SecD/F is a H⁺-driven motor, in which its

periplasmic domain (P1 head) binds a translocon-engaged substrate (*Furukawa et al., 2017*) and undergoes a conformational change coupled with the H$^+$ flow (*Tsukazaki et al., 2011*). The repetitive movements of the P1 head would allow a forward movement of a translocating chain until its release to the periplasm.

Protein translocation and membrane insertion are essential biological processes in all living organisms, and it is reasonable that factors required for the localization are tuned according to cellular demands. Bacteria have acquired unique regulatory mechanisms by which either the expression of one of duplicated translocation/integration factors is specifically induced (V.SecD2/F2 in *Vibrio alginolyticus* (*Ishii et al., 2015*) (see the next section) and YidC2 (a membrane chaperone) in *Bacillus subtilis Rubio et al., 2005*) or a unique factor is upregulated (SecA in *Escherichia coli*) (*Oliver and Beckwith, 1982*) in response to a decline in the relevant transport activity of the cell. To monitor the cellular protein translocation activities and regulate the expression of the relevant factors in real-time, these bacteria utilize a special class of nascent polypeptides, called monitoring substrates (*Ito et al., 2018*), a class of regulatory nascent polypeptide (*Tenson and Ehrenberg, 2002*), which undergoes translation elongation arrest. Currently, three proteins are known: *V. alginolyticus* VemP (*Ishii et al., 2015*), *B. subtilis* MifM (*Chiba et al., 2009*), and *E. coli* SecM (*Nakatogawa and Ito, 2001*). Translation of these proteins is arrested by the specific interaction of their intrinsic amino acid sequences, called arrest sequences or arrest motifs, with the components of the ribosome. Remarkably, the arrest motifs of different monitoring substrates are diverse, with no apparent sequence similarities.

A marine bacterium, *V. alginolyticus,* possesses two sets of the *secD/F* genes, designated *V. secD1/F1* and *V.secD2/F2,* whose products utilize Na$^+$- and presumably H$^+$-motive forces, respectively. The bacterium adapts quickly to a salinity change by replacing these SecD/F paralogues (*Ishii et al., 2015*). Although *V.secD1/F1* is expressed constitutively, the expression of *V.secD2/F2* is tightly repressed under Na$^+$-rich growth conditions, but induced under low Na$^+$ growth conditions. The *vemP* gene located upstream of *V.secD2/F2* on the same operon plays an essential role in the regulated expression of *V.secD2/F2*. VemP, a substrate of the Sec machinery, monitors the machinery's activity via its own translocation. It undergoes stable translation-arrest before the canonical termination step under a protein translocation-deficient condition. In this scenario, the stalled ribosome on a *vemP-V.secD2/F2* mRNA destabilizes a stem-loop structure in the *vemP-V.secD2* intergenic region, leading to the exposure of an otherwise masked ribosome-binding site for the *V.secD2* gene. Thus, the elongation arrest allows entry of ribosomes to that site and consequent synthesis of V.SecD2/F2. Importantly, the VemP translation-arrest occurs even under a protein translocation competent condition, but it is rapidly canceled presumably by a translocation-coupled pulling force that drives translocation of VemP.

Our previous in vivo studies showed that the majority of VemP has its signal sequence processed even in the arrested state. This strongly suggests that the arrest-cancelation of VemP occurs after its translocation has proceeded beyond the signal sequence processing event on the Sec translocon (*Mori et al., 2018*). Although this feature of VemP seems suitable for monitoring the SecD/F function, molecular mechanisms of how this is accomplished remain to be elucidated. To understand the detailed mechanism of the VemP-arrest-mediated regulation of the *V.secD2/F2* expression, it is crucial to know the dynamic interactions of VemP with other participating proteins during the arrest and its translocation-coupled cancelation processes.

Site-directed in vivo photo-crosslinking is a well-designed technique for analysis of inter- or intramolecular interactions of proteins in living cells (*Chin and Schultz, 2002*), wherein a pair of a mutated tRNA and an engineered tyrosyl-tRNA synthetase allows for in vivo incorporation of a photo-reactive amino acid analog, *p*-benzoyl-L-phenylalanine (*p*BPA), into any selected positions of a target protein by *amber* suppression (*Chin et al., 2002*). Upon UV irradiation, *p*BPA in the target protein generates a covalent crosslinking with nearby molecules. Identification of crosslinked partners provides detailed information about interactions with them at an amino acid residue level resolution, proving useful for many studies (*Miyazaki et al., 2020*). To extend this technique to acquire temporal resolution, we recently developed the PiXie (*p*ulse-chase and *in* vivo photo-*cross*linking *e*xperiment) method. This new approach entails the use of an ultra-powerful UV irradiator that shortens the UV-irradiation time so that we can combine photo-crosslinking and pulse-chase approaches. It enables us to trace dynamic in vivo folding/assembly processes of newly synthesized proteins with high temporal (in the sub-minute order) and spatial (the amino acid residue level) resolutions

(*Miyazaki et al., 2018*). Since a similar orthogonal tRNA/tRNA synthetase system for *p*BPA introduction is not available for *V. alginolyticus,* we cannot directly apply the PiXie method to the cognate bacterium to identify interactors of VemP. *Vibrio* species are closely related to enterobacteria including *E. coli* and possess a single copy of the *secA, secY, secE, secG,* and *ffh* (encoding the protein component of SRP) genes. Because each of the SecA (*Kunioka et al., 1998*), SecY (*Bhattacharyya and Das, 1997*), SecE (*Nishiyama et al., 1998*) and the two SecD/F (*Ishii et al., 2015*) homologs of *Vibrio* can function in *E. coli* cells, the basic structure and function of the Sec machinery and the interaction modes among these factors should be well conserved between these species. Therefore, photo-crosslinking analysis in *E. coli* would allow us to identify candidate proteins interacting with a VemP-nascent polypeptide in living cells.

In this study, we performed a non-biased, systematic PiXie analysis of VemP and investigated the factors that interact with a VemP-nascent polypeptide during its biosynthesis in *E. coli*. The physiological significance of the interactions observed in *E. coli* was evaluated by genetic and biochemical analysis in *Vibrio* cells. In addition, we identified an amino acid residue of VemP that imposes PpiD-dependence and thereby participates in the force-sensing regulation. Based on these analyses, we discuss how VemP integrates its translocation and arrest regulation for the real-time feedback regulation of the late step motor protein, SecD/F.

## Results

### In vivo crosslinking reveals interaction of a VemP nascent polypeptide with Ffh and PpiD as well as uL22 and translocon

VemP is a Sec-monitoring substrate in *V. alginolyticus* that monitors cellular protein secretion activity via its own translation arrest (*Ishii et al., 2015*). To identify proteins interacting with a VemP nascent polypeptide, we conducted a systematic PiXie analysis by expressing in *E. coli* a series of VemP-3xFLAG-Myc (VemP-F$_3$M) derivative having *p*BPA at various positions. We distinguished three VemP species, arrested polypeptide having unprocessed signal sequence (AP-un), arrested polypeptide whose signal sequence has been proteolytically processed (AP-pro), and the full-length mature protein (FL-m) (*Mori et al., 2018*) based on their mobilities on SDS-PAGE (*Figure 1*). The site of the *p*BPA-introduction varied from the position for the Leu-9 to the one for Phe-131. In growing cells, VemP was initially produced as the arrested forms, which were converted to FL-m with a half-life of ~0.5 min (*Mori et al., 2018*). To follow the interaction profiles in this rapid process, we used the PiXie method (*Miyazaki et al., 2018*) (See Materials and methods), which indeed revealed the *p*BPA-dependent generation of higher molecular mass bands (indicated by colored arrow-heads) indicative of the formation of crosslinked products (XLs) (*Figure 1—figure supplement 1A,B*). To identify factors physiologically interacting with VemP during its biogenesis process, we focused on XLs that gradually decreased during the chase periods (see the next section). For instance, we eliminated an XL of ~150 kDa, observed for the VemP(L40*p*BPA) variant, for the further analysis, because, while immunoprecipitatable with an anti-SecA antibody (*Figure 1—figure supplement 2A*), it remained stable during the chase periods, in sharp contrast to other XLs observed for VemP derivatives having a *p*BPA at position either 56 or 124 (*Figure 1—figure supplement 2B* and see the next section). The XL of ~150 kDa would represent the adduct between SecA and VemP, but not reflect physiological interactions that occur during translocation of VemP.

An XL of ~30 kDa, observed for the VemP(W124*p*BPA) variant, was immunoprecipitatable with an anti-uL22 antibody (*Figure 1B*, *right*). The proximity of Trp-124 to uL22 in vivo is consistent with a recent cryo-EM study showing that Trp-124 of arrested VemP is positioned close to the ribosomal protein uL22 (*Figure 1B*, *left*) (*Su et al., 2017*). VemP(F81*p*BPA) and VemP(T83*p*BPA) generated XLs of ~25 kDa. We envisioned that the crosslinked partner could either be SecE (~14 kDa) or SecG (~12 kDa), small components of the translocon, because VemP engages in the Sec-dependent translocation (*Mori et al., 2018*). We reasoned that these XLs contained SecG because they reacted with an anti-SecG antibody (*Figure 1C* and *Figure 1—figure supplement 3A*) and disappeared in the Δ*secG* strain (VemP(F81*p*BPA); *Figure 1—figure supplement 3A*). To investigate possible crosslinking with SecY, the main component of the translocon, we used a host strain having a chromosomal *secY-his*$_{10}$ gene. We detected the SecYxVemP XLs for the VemP derivatives having *p*BPA in a middle region (Arg-74, Asp-76, Leu-78, Asn-82, and Trp-86) of VemP among materials pulled-down by polyhistidine

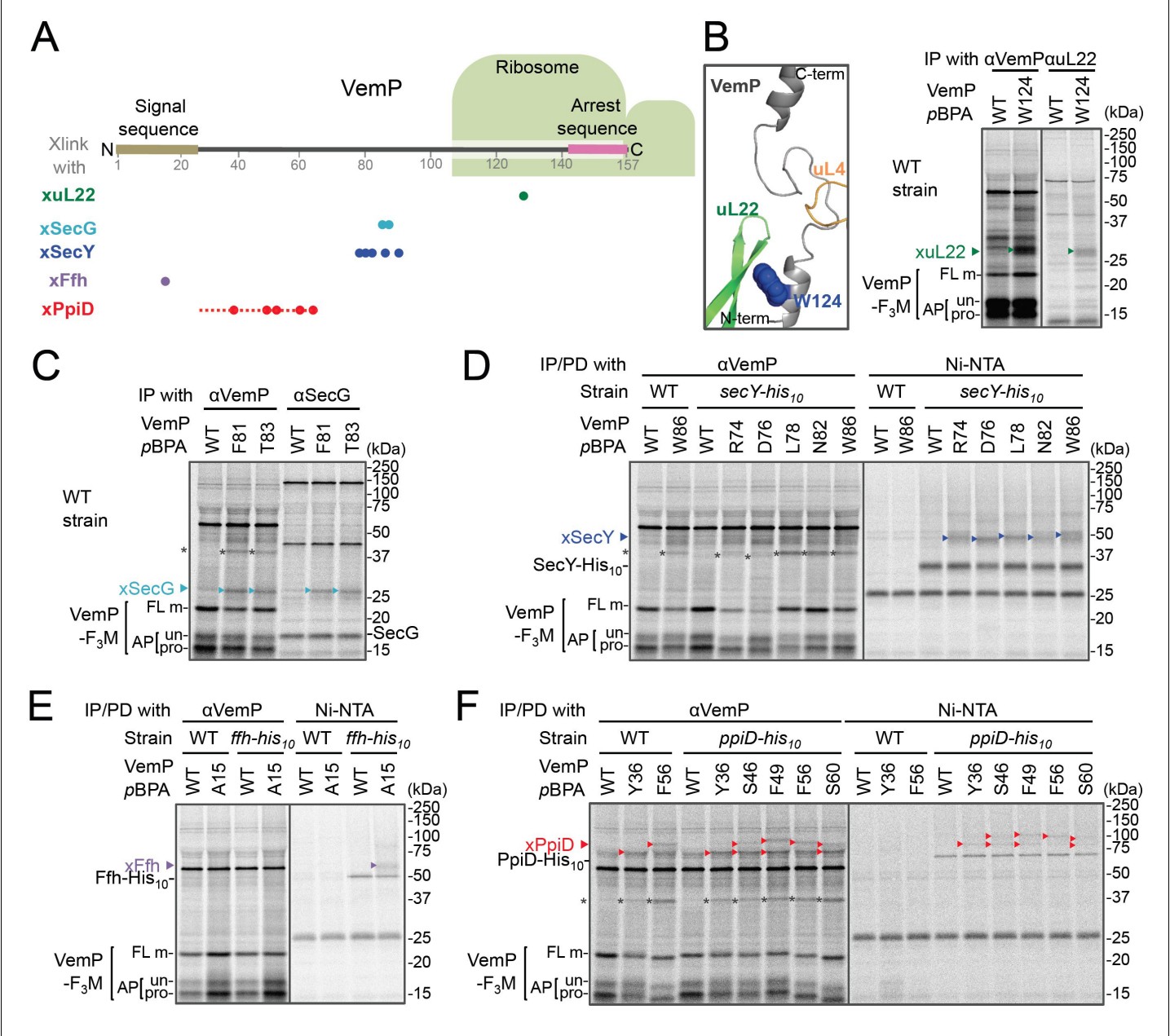

**Figure 1.** In vivo crosslinking reveals interaction of a VemP nascent polypeptide with Ffh and PpiD as well as uL22 and translocon. (**A**) Summary of proteins crosslinked with VemP in the arrested state. Colored circles represent the positions at which crosslinking with the indicated proteins was observed in B–F. Dashed red line shows the positions at which cross-linking with PpiD was detected in *Figure 1—figure supplement 1*. (**B–F**) PiXie analysis of VemP. Wild type *E. coli* cells were used in **B** and **C**. *E. coli* cells expressing SecY-His$_{10}$, Ffh-His$_{10}$, or PpiD-His$_{10}$ from the chromosome were used in **D**, **E** and **F**, respectively. Wild type cells were used as negative controls in **D**, **E** and **F**. Cells were grown, induced to express a VemP-F$_3$M derivative and pulse-labeled with [$^{35}$S]Met for 30 s, followed by 30 s-chase as described in Materials and methods. Cells were UV-irradiated for 1 s and immediately acid-treated. Labeled proteins were subjected to immunoprecipitation (IP) or pull-down (PD) with Ni-NTA agarose, separated by SDS-PAGE and analyzed by phosphorimaging. The result shown is a representative of two independent experiments that were conducted using the same transformants (i.e. two technical replicates). Asterisks represent crosslinked dimers of VemP-F$_3$M. An enlarged view around Trp-124 of VemP in a VemP-stalled ribosome complex (PDB:5nwy; *Su et al., 2017*) is shown in **B**, *left*.

The online version of this article includes the following figure supplement(s) for figure 1:

**Figure supplement 1.** Systematic PiXie analysis of a nascent VemP.

**Figure supplement 2.** Identification and characterization of VemPxSecA crosslinked products.

**Figure supplement 3.** Identification of VemP crosslinked products.

**Figure supplement 4.** Identification of VemP crosslinked products using His-tagged derivatives of candidate proteins.

*Figure 1 continued on next page*

*Figure 1 continued*

**Figure supplement 5.** Immunoprecipitation of VemP-crosslinked products using anti-FLAG antibodies.

affinity isolation (*Figure 1D* and *Figure 1—figure supplement 4A*). Thus, the middle region of the newly synthesized VemP chain contacts the translocon.

For this interaction to occur, VemP might use some targeting mechanism that delivers it to the translocon. SecM, the Sec-monitoring substrate in *E. coli* was suggested to depend on the SRP pathway (*Nakatogawa and Ito, 2001*), which consists of the 4.5S RNA and the signal sequence-recognition subunit, Ffh. We found that VemP also interacts with SRP through the PiXie analysis using a strain carrying chromosomal *ffh-his$_{10}$*. Placement of *p*BPA within the signal sequence (Ala-15 position) allowed for the generation of a VemPxFfh XL that was His-tag affinity isolated (*Figure 1E* and *Figure 1—figure supplement 4B*). We suggest that SRP recognizes the signal sequence of VemP.

Placement of *p*BPA at the C-terminal vicinity of the signal sequence (residue 30 to 60) led to the generation of XLs with apparent molecular sizes of ~75–100 kDa (*Figure 1—figure supplement 1A, B*). As partner candidates, we highlighted YidC (~60 kDa) (*Kuhn et al., 2017*), SecD (~68 kDa) (*Tsukazaki, 2018*), and PpiD (~69 kDa) (an inner membrane-anchored periplasmic chaperone) (*Antonoaea et al., 2008*) based on their sizes and their involvement in protein translocation. We addressed the possibility of their involvement using their His-tagged derivatives and found that a XL was His-tag affinity-isolated when a cell with chromosomal *ppiD-his$_{10}$* was used (*Figure 1F*), but not with the other *his$_{10}$* constructs (*Figure 1—figure supplement 4C,D*). Furthermore, the XL was immunoprecipitated with an anti-PpiD antibody and were not generated in a Δ*ppiD* cell (*Figure 1—figure supplement 3B*). These results indicate that PpiD interacts with the VemP region (Tyr-36 to Ser-60) that follows the periplasmic end of the signal sequence.

None of the VemP XLs with uL22, Ffh, SecG, SecY, and PpiD described above were immunoprecipitated with the anti-FLAG antibody (*Figure 1—figure supplement 5*), indicating that the VemP component in these XLs lacks the C-terminal tag and, therefore, are in the elongation-arrested state. It follows then that each of the partner proteins binds to the nascent VemP'-tRNA tethered to the ribosome. In addition to the above XLs, we observed ~37 kDa XLs with a number of VemP*p*BPA derivatives (*Figure 1—figure supplement 1*), some of which proved to be immunoprecipitable with the anti-FLAG antibody (*Figure 1—figure supplement 5*). We interpret that these ~37 kDa XLs represent dimer forms of FL-m generated in the periplasm. Thus, we did not analyze these XLs any further. Although some of other XLs shown in *Figure 1—figure supplement 1* could represent VemP-crosslinking with still un-identified cellular factors involved in the arrest/cancelation process of VemP, we did not conduct further analysis in this study. *Figure 1A* summarizes the crosslinking features of nascent VemP polypeptide.

## The VemP nascent polypeptide interacts sequentially with uL22/Ffh, the translocon, and PpiD

To gain insight into the timing and order of the molecular interactions involving VemP, we next examined in vivo stability of the XLs during the chase period using appropriate *p*BPA variants. The intensities of the radioactivities associated with the arrested and non-crosslinked VemP decreased during the chase period due to the secretion-coupled arrest cancelation. While XLs intensities also declined during the chase (*Figure 2A,B* and *Figure 2—figure supplement 1*), a careful comparison of the decrease rates showed that the kinetics of the XLs decrease did not always coincide exactly with that of the arrested-VemP. While the decrease rates of the SecG-XLs and the arrested VemP were almost the same, the uL22- and Ffh-XLs decreased slightly faster than the arrested VemP (*Figure 2B,C* and *Figure 2—figure supplement 2*). These results suggest that Ffh (SRP) and uL22 interact with the arrested VemP polypeptide before targeting to the translocon, in agreement with their cellular functions and localization. This was verified by examining the effect of NaN$_3$, a SecA inhibitor, on the VemP-crosslinkings. NaN$_3$ treatment of cells causes the stabilization of the AP-un form of VemP in living cells (*Mori et al., 2018*), suggesting that inhibition of SecA prevents VemP from initiating its translocation into the translocon. We confirmed it by pulse-chase experiments using a *secA51* (Ts) mutant cells (*Figure 2—figure supplement 3*). As expected, the arrest cancelation of VemP was severely and specifically retarded at high temperature in the *secA51*(Ts) mutant

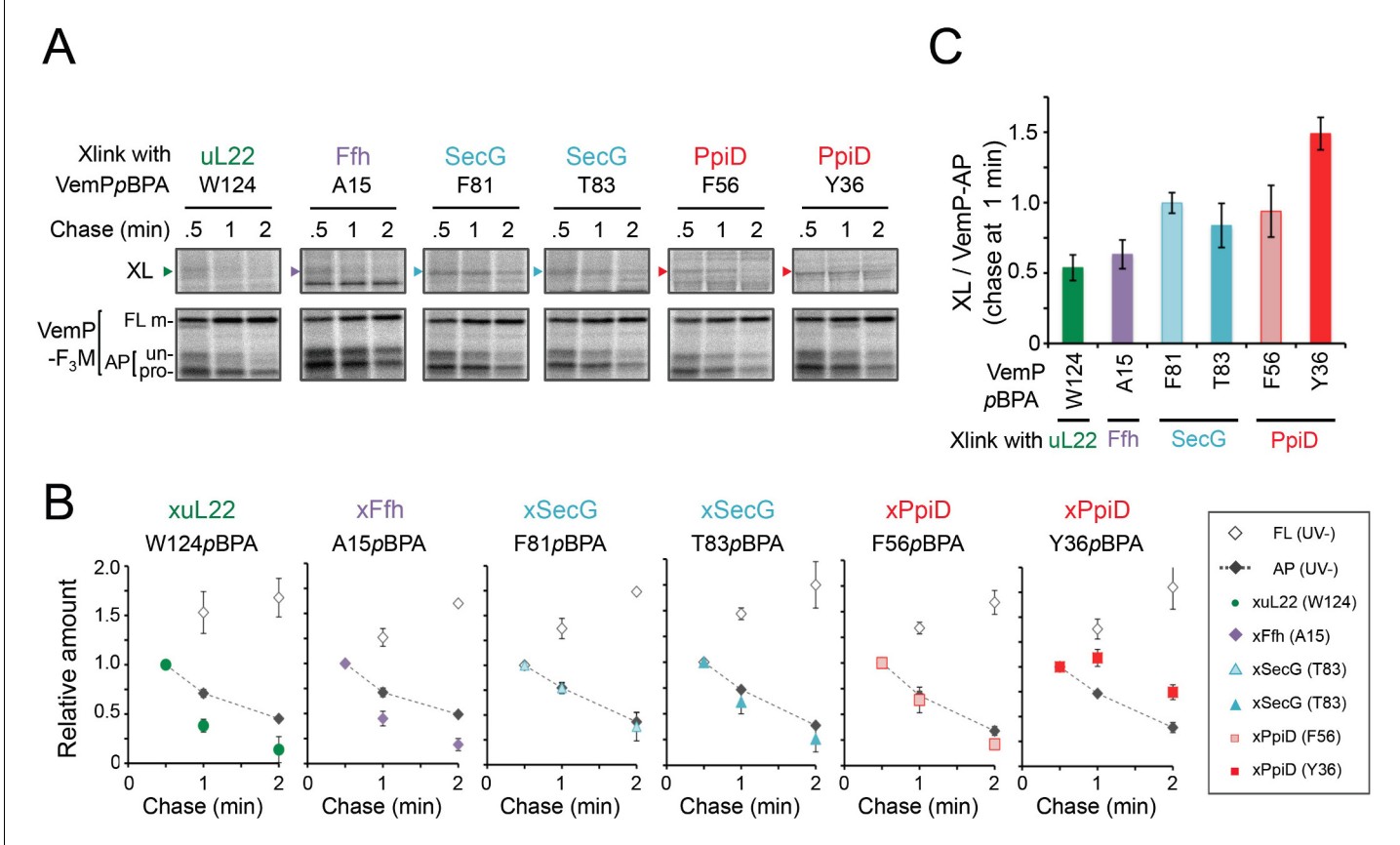

**Figure 2.** The VemP nascent polypeptide interacts sequentially with uL22/Ffh, the translocon, and PpiD. (A) PiXie analysis of VemP. Cells were grown, induced to express a VemP derivative with pBPA, pulse-labeled and chased as in *Figure 1*. At the indicated time points in the chase period, cells were UV-irradiated for 1 s and then acid-treated. Labeled proteins were subjected to IP with an anti-VemP antibody or pull-down with Ni-NTA agarose. The same strains were parallelly pulse-labeled, chased, and subjected to IP with the anti-VemP antibody without the UV irradiation. Isolated proteins were analyzed as in *Figure 1*. Portions of the gel images showing the XLs (*upper gels*) or showing the VemP-derived bands of the UV-unirradiated samples (*lower gels*) are presented. Full images of the gels for all the results are presented in *Figure 2—figure supplement 1B*. The result shown is a representative of three biological replicates. (B) Relative amounts of the VemP arrested forms and the VemP-XLs. The band intensity of VemP-FL (open diamonds), VemP-APs (AP-un + AP-pro) (closed diamonds), and XLs (colored symbols) in A was quantitated and the mean values of the relative values (the value at the 0.5 min was set to 1) were plotted (error bars are S.D.; (n = 3)). The detail procedure for quantification of immunoprecipitated bands in a representative result is presented in *Figure 2—figure supplement 2*. (C) Relative crosslinking efficiency of the arrested VemP. The values for the relative intensities of XLs at 1 min to the corresponding average intensities of VemP-APs were calculated from the results in B. The mean values are shown with S.D. (n = 3 biological replicates). See *Figure 2—source data 1* for gel images and quantitated band intensities data for A–C.

The online version of this article includes the following source data and figure supplement(s) for figure 2:

**Source data 1.** Zip file containing gel images and quantified band intensity data for the PiXie experiments.
**Figure supplement 1.** PiXie analysis of VemP interactions.
**Figure supplement 1—source data 1.** Zip file containing gel images and quantified band intensity data for the pulse-chase experiments.
**Figure supplement 2.** The procedure of the quantification of immunoprecipitated bands.
**Figure supplement 3.** Effects of the *secA51* mutation on the stability of the arrested VemP.
**Figure supplement 3—source data 1.** Zip file containing gel images and quantified band intensity data for the pulse-chase experiments.
**Figure supplement 4.** Effects of $NaN_3$ treatment on VemP crosslinking.
**Figure supplement 4—source data 1.** Zip file containing gel images and quantified band intensity data for the PiXie experiments.

cells, but not in its isogenic wild type cells. We found that the $NaN_3$ treatment increased the VemP crosslinking with uL22 and Ffh, but decreased its crosslinking with SecY, SecG, and PpiD (*Figure 2— figure supplement 4A,B*). These results suggest that the arrested VemP interacts with SRP and uL22 before the interactions with the translocon and PpiD on the membrane. In contrast to the uL22- and Ffh-XLs, the PpiD-XLs tended to decrease more slowly than SecG-XLs or the arrested VemP

(*Figure 2B,C*). This suggests that VemP is recognized by PpiD after it engages with the translocon, being consistent with the topology of PpiD that has a functional domain in the periplasm.

## Ffh (SRP) functions in the targeting of VemP

We examined the role of Ffh (SRP) in the membrane translocation and arrest-cancelation of VemP by pulse-chase experiments using an *E. coli* strain, WAM121, in which *ffh* is transcribed from the arabinose promoter (*de Gier et al., 1996*). The Ffh-depletion upon removal of arabinose from the medium stabilized the AP-un form of VemP (*Figure 3A*), indicating an impairment of the targeting of the VemP-ribosome complex to the translocon. Consistent with the observation, deletion of the *secB* gene that encodes a secretion-specific chaperone severely compromised export of MBP, a SecB-dependent substrate, but exhibited the normal arrest cancelation of VemP (*Figure 3—figure supplement 1*), supporting the co-translational targeting of a VemP-ribosome complex. The prolonged translation-arrest of VemP is expected to induce the expression of the downstream *V.secD2/F2* genes (*Ishii et al., 2015*). To verify this point, we examined the V.SecD2 expression in WAM121 cells carrying a *vemP-V.secD2/F2* plasmid, and found that the synthesis of V.SecD2 was indeed elevated under the Ffh-depleted conditions (*Figure 3B*). We also examined the effect of an Ffh-depletion on the V.SecD2 expression in the cognate organism, *V. alginolyticus* (*Figure 3C*). Under a $Na^+$-rich growth condition, the $Na^+$-driven V.SecD1/F1 in this organism was fully functional, leading to the efficient arrest cancelation of VemP and consequent tight repression of V.SecD2 (*Ishii et al., 2015*). In the Ffh-depletable strain of *V. alginolyticus* with the arabinose promoter-controlled *ffh*, the V.SecD2 expression was induced in the arabinose-free medium even in the presence of a sufficient level of $Na^+$, consistent with the notion that the Ffh-depletion compromises the targeting of VemP and stabilizes its translation arrest. These results suggest that SRP is crucial for the translocon targeting of VemP, which is a prerequisite for the proper regulation of the V.SecD2/F2 expression in *Vibrio*.

## PpiD and SecD/F cooperate to facilitate the translocation and the arrest cancelation of VemP

We then carried out functional studies of PpiD, identified in this study as a crosslinking partner of VemP. Pulse-chase analysis showed that VemP was kept in the arrested state much more stably in the Δ*ppiD* cells than in the *ppiD*+ *E. coli* cells (*Figure 4A*). We note that translocation of a secretory protein MBP (maltose-binding protein) was also retarded by the *ppiD* disruption, albeit less pronouncedly (*Figure 4A*). The loss of *ppiD* increased the V.SecD2 expression in *E. coli* (*Figure 4B*) and in *Vibrio* (*Figure 4C*). These results show that PpiD plays a crucial role in the arrest-cancelation of VemP. The PpiD disruption caused an accumulation of the AP-pro form of VemP in contrast to the Ffh depletion that led to the AP-un accumulation (compare *Figures 3A* and *4A*). These results support the notion that PpiD acts on the periplasmic side after the signal sequence cleavage of the arrested VemP, whereas Ffh acts earlier on the cytosolic side. Our kinetic analysis already showed that PpiD is crosslinked to VemP at a late step of its translocation processes (*Figure 2*).

Next, we addressed how PpiD participates in the arrest cancelation of VemP, whose role is to regulate the expression of SecD/F that facilitates a late step of translocation (*Mori et al., 2018*). The role of SecD/F itself in the VemP arrest cancelation can be seen from the prolonged VemP arrest in the *secD1* (a mutation causing SecD/F depletion) *E. coli* cells (*Figure 4D*, vector lanes) and the enhanced arrest cancelation by overproduction of SecD/F (*Figure 4D*, *left*, vector vs. *secD/F*). As both SecD/F and PpiD are involved in later steps of VemP translocation, they might cooperate in canceling the elongation arrest. Alternatively, they might have redundant functions. The arrested state of VemP was stabilized in the Δ*ppiD* cells (*Figure 4D*, vector lanes), showing that its function is required for the active translocation and the arrest cancelation of VemP. Whereas the overproduction of PpiD facilitated the arrest cancelation, as expected, SecD/F's overproduction was ineffective in relieving the arrested state due to the *ppiD* defect. Conversely, PpiD overproduction was ineffective in relieving the arrested state due to the *secD1* mutation (*Figure 4D*). These results rule out the possibility that the roles of SecD/F and PpiD in arrest cancelation are redundant. Instead, both of them are specifically required for the cancelation of VemP arrest, raising the possibility that they interact with each other. We then studied this point by in vivo photo-crosslinking. When *p*BPA was incorporated into the position of Asp-359 in the P1 head of SecD, its XLs with PpiD were detected

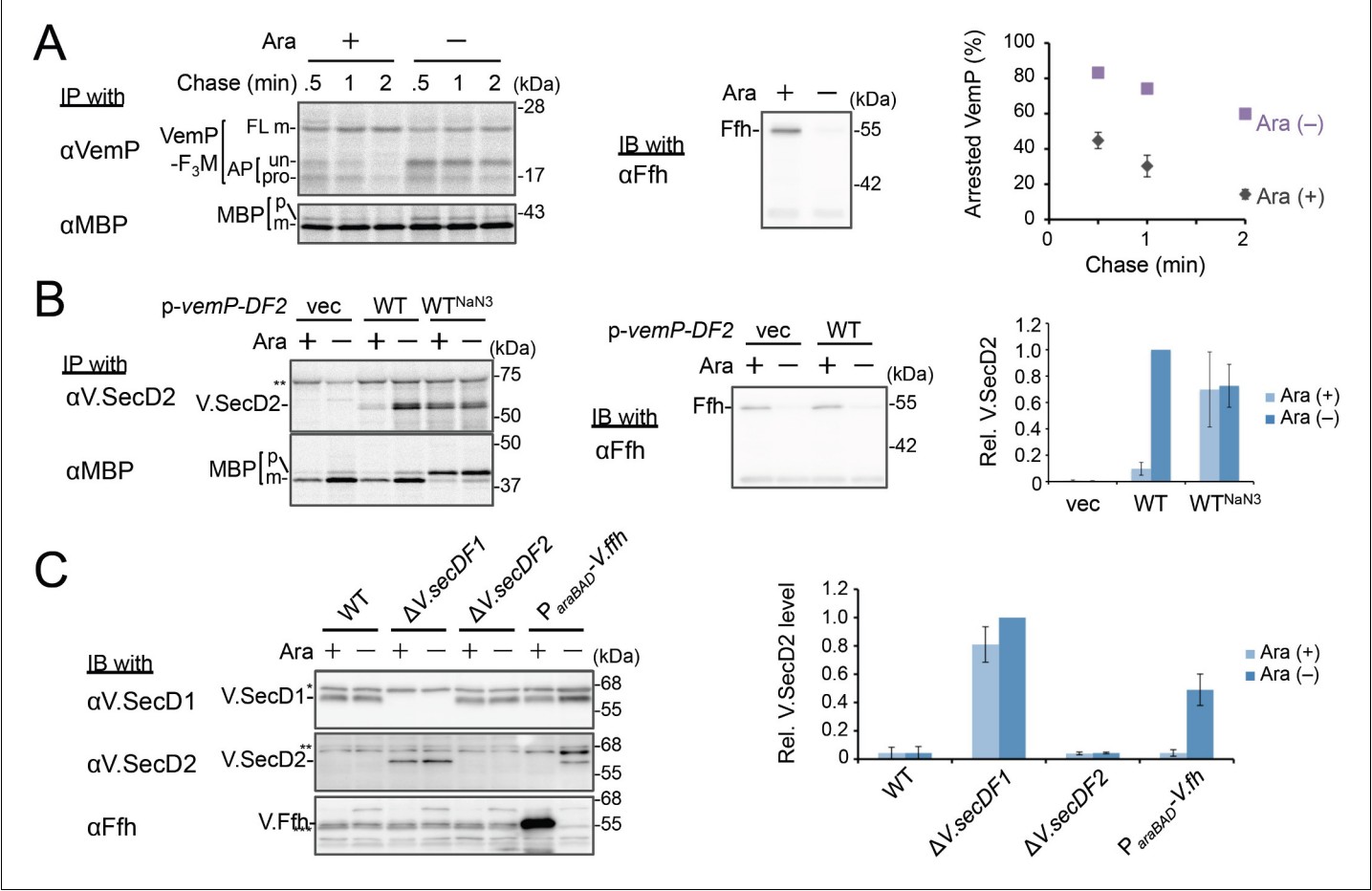

**Figure 3.** Ffh (SRP) functions in the targeting of VemP. (**A**) Effects of Ffh-depletion on stability of the arrested-VemP. (*left*) Ffh-depletable cells carrying a *vemP-f₃m* plasmid were grown in the M9 medium with (+) or without (–) 0.05% arabinose, induced, pulse-labeled and chased as in *Figure 1*. At the indicated chase time points, total cellular proteins were acid-precipitated, subjected to IP and analyzed as in *Figure 1*. (*middle*) In parallel, a portion of the cultures just before pulse-labeling was subjected to immunoblotting (IB) analysis with an anti-Ffh antibody. (*right*) Percentages of the arrested VemP in *left* were calculated by the equation described in Materials and methods. Values are means ± S.D. (n = 3 technical replicates). (**B**) Effects of Ffh-depletion on the expression of *V.secD2*. (*left*) The Ffh-depletable cells carrying an empty vector or a *vemP-V.secD2/F2* plasmid (WT) were grown, induced, and pulse-labeled for 1 min. Total cellular proteins were acid-precipitated. For the WT^NaN3 samples, the cells carrying the *vemP-V.secD2/F2* plasmid were pretreated with 0.02% NaN₃ for 5 min before pulse-labeling. Labeled proteins were subjected to IP, and analyzed as in *Figure 1*. (*middle*) In parallel, a portion of the cultures just before the pulse-labeling was subjected to IB with the anti-Ffh antibody. The intensity of the V.SecD2 band from each lane was quantitated. Values are means ± S.D. (n = 3 technical replicates) (the value for WT in the presence of arabinose was set to 1). (**C**) Effects of Ffh-depletion on the expression of *V.secD2* in a *Vibrio* cell. (*left*) The *Vibrio* cells indicated were grown at 30°C in VC-medium with (+) or without (–) 0.2% arabinose for 3 hr. Total cellular proteins were acid-precipitated, and analyzed by IB. (*right*) The intensity of the V.SecD2 band from each lane was quantitated. Values are means ± S.D. (n = 3 technical replicates) (the value for the *ΔV.secD1/F1* cells in the presence of arabinose was set to 1). The asterisks (*, **, ***) in **B** and, **C** represent un-related proteins recognized by the indicated antibodies. See *Figure 3—source data 1* for gel images and quantitated band intensities data for **A–C**.

The online version of this article includes the following source data and figure supplement(s) for figure 3:

**Source data 1.** Zip file containing gel images.
**Figure supplement 1.** Effects of a *secB*-deletion on the stability of the arrested VemP.

(*Figure 4E*). Thus, PpiD resides in close proximity to the mobile P1 head of SecD of the VemP-translocating translocon. Generation of the two XLs could represent crosslinking of *p*BPA at position 359 in SecD to two different positions in PpiD.

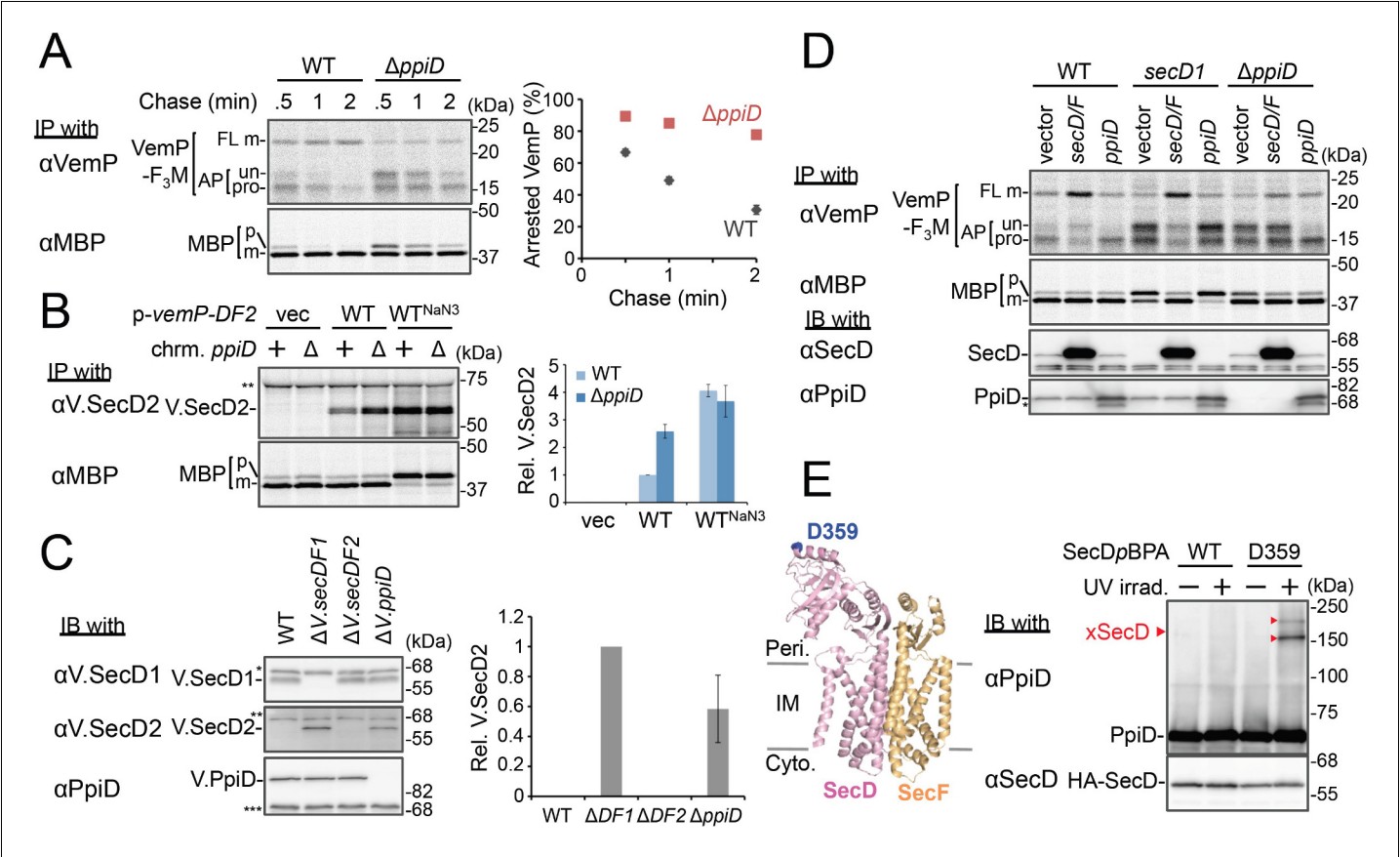

**Figure 4.** PpiD and SecD/F cooperate to facilitate the translocation and the arrest cancelation of VemP. (A) Effects of the *ppiD* disruption on the stability of the arrested-VemP. (*left*) WT cells or Δ*ppiD* cells carrying the *vemP-f₃m* plasmid were grown, induced, pulse-labeled, and chased as in *Figure 1*. At the indicated chase time points, total cellular proteins were acid-precipitated and subjected to IP and analyzed as in *Figure 1*. (*right*) The mean values of arrested VemP (%) are plotted with S.D. (n = 3 technical replicates). (B) Effects of the *ppiD* disruption on the expression of *V.secD2*. (*left*) WT cells or Δ*ppiD* cells carrying either an empty vector or a *vemP-V.secD2/F2* plasmid (WT) were grown, induced as in **A** and pulse-labeled for 1 min. For the WT^NaN3 samples, the cells carrying the *vemP-V.secD2/F2* plasmid were pretreated with 0.02% NaN₃ for 5 min before pulse-labeling. Labeled proteins were subjected to IP and analyzed as in *Figure 1*. The intensities of the V.SecD2 bands were quantitated. Values are means ± S.D. (n = 3 technical replicates) (the value for the WT cells was set to 1). (C) Effects of the *ppiD* disruption on the expression of *V.secD2* in *Vibrio* cells. (*left*) The indicated *Vibrio* cells were grown at 30℃ in the VC-medium for 2 hr. Total cellular proteins were acid-precipitated and subjected to IB. (*right*) The intensities of the V.SecD2 bands were quantitated. Values are means ± S.D. (n = 3 technical replicates) (the value for the Δ*V.secD1/F1* cells was set to 1). The asterisks (*, **, ***) in **B** and **C** represent un-related proteins recognized by the indicated antibodies. (D) Roles of SecD/F and PpiD in the arrest-release of VemP. A pRM83c-based plasmid carrying either *his₁₀-secD/F* or *ppiD*, or the empty vector was introduced, in addition to pHM1021-*vemP-f₃m*, into wild type cells and cells having either the *secD1* or the Δ*ppiD* mutation. These cells were grown at 37℃ for 2.5 hr as in *Figure 1*. A half of the cell cultures were removed and acid-treated. Precipitated proteins were subjected to IB (*lower two panels*). The remaining cells were induced with 1 mM IPTG for 15 min, pulse-labeled for 30 s and chased for 30 s. Acid-precipitated proteins were subjected to IP and analyzed as in *Figure 1* (*upper two panels*). The result shown is a representative of two technical replicates. The asterisk in the lowest gel represents a degradation product of PpiD. (E) In vivo photo-crosslinking of SecD with PpiD. Cells were grown in L medium containing 0.5 mM *p*BPA until early log phase at 37℃ and induced with 0.02% arabinose for 1 hr to express the indicated SecD/F variants. The cultures were divided into two portions, each of which was treated with or without UV-irradiation for 10 min at 4℃. Total cellular proteins were acid-precipitated and subjected to IB. The result shown is a representative of two technical replicates. A crystal structure of SecD/F (PDB:3aqp; *Tsukazaki et al., 2011*) is shown in the *left*. See *Figure 4—source data 1* for gel images and quantitated band intensities data for **A**–**C**.

The online version of this article includes the following source data for figure 4:

**Source data 1.** Zip file containing gel images.

## Conserved Arg-85 has a role in the regulation of secretion-coupling of the VemP arrest cancelation

Our systematic *p*BPA scanning incidentally revealed that incorporation of this amino acid analog to some VemP residues compromised the translation arrest of VemP (*Figure 1—figure supplement 1A*

and *Figure 5—figure supplement 1A*). We selected several residues among them (shown in *Figure 5—figure supplement 1B*), and replaced them individually with Trp (having a bulky, hydrophobic side chain like *p*BPA). The Trp substitutions destabilized the translation arrest in similar ways to the corresponding *p*BPA substitutions. Among them, the R85W mutation had the strongest effect; it decreased the ratio of arrested forms to total VemP to the level comparable with that observed with a previously identified arrest motif mutation, W143A (*Mori et al., 2018*). Several other replacements of Arg-85 by residues of different properties also impaired apparently the translation arrest of VemP (*Figure 5A*). The R85W mutation abrogated the arrest-mediated up-regulation of *V.secD2* (*Figure 5—figure supplement 1C*). These results showed that Arg-85 is crucial for the stable translation arrest of VemP in vivo.

We conceived two possibilities that account for the observed phenotypes of the Arg-85 mutations. First, this residue belongs to the arrest sequence, although it presumably resides outside of the ribosome in the arrested translation complex. In this case, the Arg-85 mutations simply lead to a loss of arrest proficiency. Secondly, Arg-85 regulates the arrest proficiency by partially antagonizing the secretion-coupled arrest cancelation. In this scenario, the Arg-85 mutation may sensitize the VemP nascent chain to the secretion-generated pulling force and consequently lead to the apparent arrest-defective phenotype in vivo. To distinguish these possibilities, we examined the VemP(R85W)

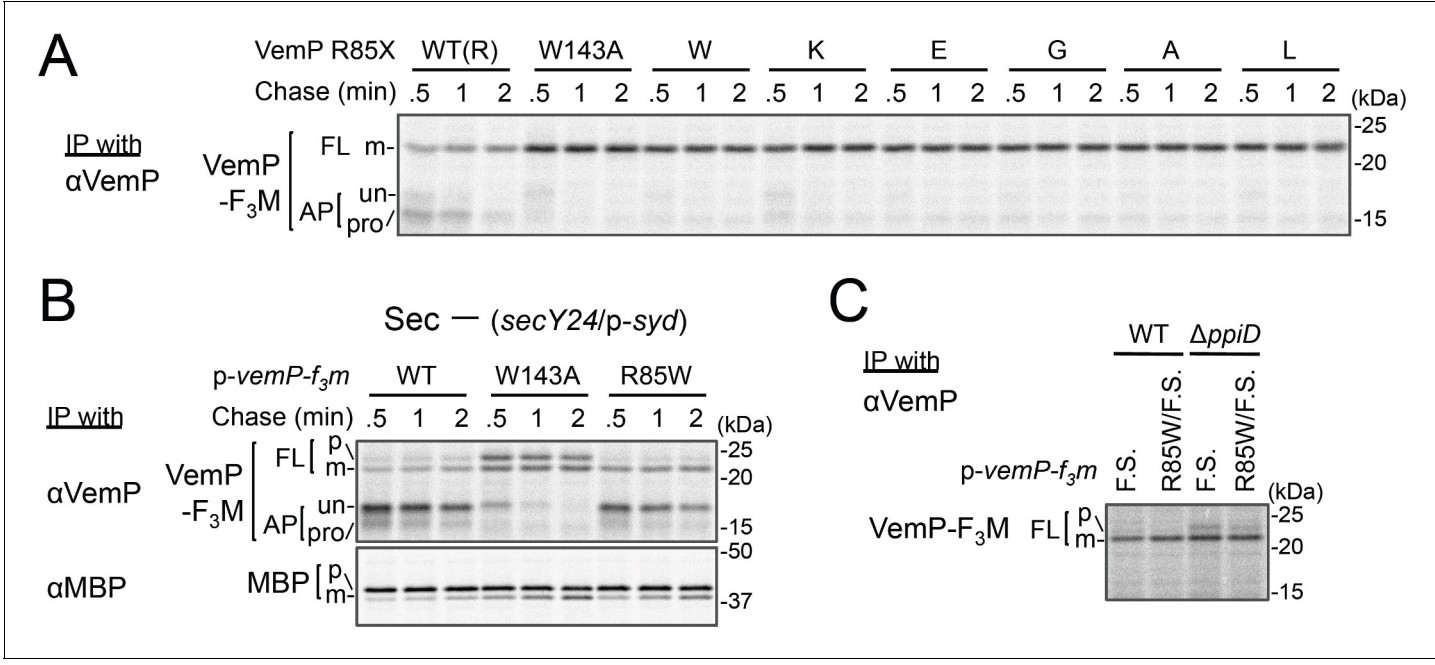

**Figure 5.** Conserved Arg-85 has a role in the regulation of secretion-coupling of the VemP arrest cancelation. (**A**) Effects of Arg-85 mutations on the stability of the arrested VemP in vivo. Cells were grown, induced to express the indicated VemP-F₃M derivatives, and used for the pulse-chase experiments as in *Figure 4A*. Labeled proteins were subjected to IP with an anti-VemP antibody and analyzed as in *Figure 1*. The result shown is a representative of three technical replicates. (**B**) Behavior of the VemP(R85W) mutant under Sec-deficient conditions. Cells defective in protein translocation (*secY24/pSTV28-syd*) expressing the indicated VemP-F₃M derivatives were examined by the pulse-chase experiment as described in *Figure 4A* except that the export of MBP was also examined by IP with an anti-MBP antibody. The result shown is a representative of two technical replicates. (**C**) Effects of the R85W mutation on translocation of the VemP(F.S.) derivative that undergoes no translation arrest. WT cells or Δ*ppiD* cells were grown, induced to express the indicated VemP(F.S.)-F₃M derivatives, and used for the pulse-chase experiments as in *Figure 4A*. Labeled proteins were subjected to IP with an anti-VemP antibody and analyzed as in *Figure 1*. The result shown is a representative of three biological replicates. The online version of this article includes the following source data and figure supplement(s) for figure 5:

**Figure supplement 1.** The conserved Arg-85 residue is important for the stability of the arrested VemP in vivo.

**Figure supplement 1—source data 1.** Zip file containing gel images and quantified band intensity data for the pulse-labeling experiments.

**Figure supplement 2.** Effects of a *ppiD* deletion on translocation and arrest cancelation of the VemP variant Cells were grown, induced to express the indicated VemP-F3M derivative, and pulse-labeled as in *Figure 4*.

**Figure supplement 2—source data 1.** Zip file containing gel images and quantified band intensity data for the pulse- labeling experiments.

**Figure supplement 3.** Sequence alignment of VemP orthologues.

mutant protein behaviors under secretion-defective conditions. In the Sec-deficient cells (see Materials and methods), the arrested form of WT VemP was stabilized (*Mori et al., 2018*), while that of VemP(W143A) was rapidly converted to the full-length form (*Figure 5B*). In contrast, the R85W mutant did not exert a negative effect on the arrest in the secretion-impaired cell (*Figure 5B*). As another means to render VemP indifferent to secretion, we deleted its signal sequence. The R85 mutation did not affect the elongation arrest in the absence of the signal sequence. By sharp contrast, the W143A mutation abolished the arrest even in the absence of the signal sequence (*Figure 5—figure supplement 1D*). Thus, the mode of involvement of Arg-85 in the elongation arrest is fundamentally different from that of Trp-143.

These data revealed that Arg-85 acts after targeting of the VemP-ribosome complex to the translocon, probably during its translocation. To examine the role of Arg-85 in the SecD/PpiD-dependent translocation of VemP, we examined the effects of the R85W and W143A mutation of VemP, individually or in combination on translocation of VemP in wild type and Δ*ppiD* strains (*Figure 5—figure supplement 2*). In the Δ*ppiD* strain, while a significant fraction of the VemP(W143A) was retained in the arrested forms, the VemP(R85W/W143A) double mutant was almost completely translocated to generate the mature (processed) FL form. The results suggest that the R85W mutation allows the efficient translocation of VemP even in the absence of PpiD, indicating that the mutation at Arg-85 reduces PpiD-dependency of the VemP translocation. To evaluate the role of Arg-85 in the VemP translocation without the elongation arrest, we used the VemP(F.S.) mutant, in which the arrest motif sequence had been replaced with a completely different amino acid sequence by a frame shift (F.S.) mutation (*Mori et al., 2018*), and assayed the VemP translocation in the wild type and the Δ*ppiD* strains by pulse-labeling (*Figure 5C*). As expected, the VemP(F.S.) derivative no longer underwent the elongation arrest even in the Δ*ppiD* strain. We observed a small but significant amount of the full-length precursor form of VemP(F.S.) in the Δ*ppiD* strain, suggesting the translocation of VemP(F.S.) depends on PpiD. Interestingly, the additional introduction of the R85W mutation into VemP(F.S.) greatly reduced the amount of the precursor. These results indicate that while the Arg-85 residue somehow retards the translocation of VemP independently of its translation arrest, it is released by the function of PpiD; in other words, Arg-85 imposes PpiD-dependency on the translocation of VemP. The function of Arg-85 could repress the arrest cancelation of VemP until translocation of VemP proceeds to an appropriate stage, at which PpiD and SecD/F come in to enhance the late stage of translocation.

## Discussion

VemP in *Vibrio* controls the SecD/F expression through its force-sensitive translation arrest. Although we previously suggested that the arrest cancelation of VemP is coupled to a late step of translocation (*Mori et al., 2018*), the details of the responsible force and how this regulatory process occurs remain largely unknown. To understand the molecular mechanisms that lead to the elongation arrest and its cancelation, we performed a systematic PiXie analysis of VemP, which allowed us to identify cellular factors that sequentially interact with the newly synthesized VemP polypeptide chain. We previously used the PiXie method, which allows precisely time-controlled and residue-specific crosslinking, to analyze the post-translational maturation processes of protein complexes with different cellular localizations in bacteria (*Miyazaki et al., 2018*). This method should also be useful for studying co-translational events (*Kramer et al., 2019*; *Pechmann et al., 2013*) that a nascent polypeptide experiences during translation. In particular, a class of polypeptides that undergoes translation arrest would be suitable targets of the PiXie analysis. The molecular interaction behaviors of such arrest peptides, including bacterial VemP, SecM, MifM, and mammalian XBP1u (*Chiba et al., 2009*; *Ishii et al., 2015*; *Nakatogawa and Ito, 2002*; *Yanagitani et al., 2011*), would be highly relevant for our understanding of these nascent polypeptides in the cell. Thus, in this work, we applied the PiXie method to VemP, which we have identified and characterized previously (*Ishii et al., 2015*).

We identified Ffh and PpiD as factors directly interacting with the arrested VemP in addition to the ribosomal protein uL22 and the translocon components (*Figure 1*). The kinetic analyses suggest their sequential interactions with VemP (*Figure 2*); VemP initially interacts with uL22 and the cytosolic Ffh, then with the membrane-embedded translocon, and lastly with PpiD, a periplasmically exposed chaperone. The crosslinking of VemP with Ffh and PpiD should have functional significance rather than representing a result of a non-specific collision of VemP with these proteins, as

suggested by our genetic analyses using the Ffh-depletable and the *ppiD* deletion strains. The observation that the distinct regions in VemP interact with uL22, Ffh, the translocon, and PpiD (*Figures 1* and *6A*, *left*) is consistent with pre-targeting interactions of the arrested VemP with uL22 and Ffh and a post-targeting interaction with PpiD (*Figure 6A*, *right*) (see Appendix1: Interactions of a VemP nascent chain with Sec translocon and the related components). A cryo-EM study showed that a VemP nascent polypeptide in the ribosomal exit tunnel forms a compacted secondary structure (*Su et al., 2017*). In this structure, Trp-124 of VemP contacts to the uL22 protein. The observed crosslinking of *p*BPA at Trp-124 with uL22 is consistent with the reported structure. Interestingly, we found that the crosslinking representing the VemP-uL22 interaction was enhanced markedly when cells were treated with $NaN_3$ that should have compromised an early step of the VemP translocation by inhibiting SecA. Thus, the VemP's interaction with uL22 occurs before its initial insertion into the translocon (*Figure 2—figure supplement 4*).

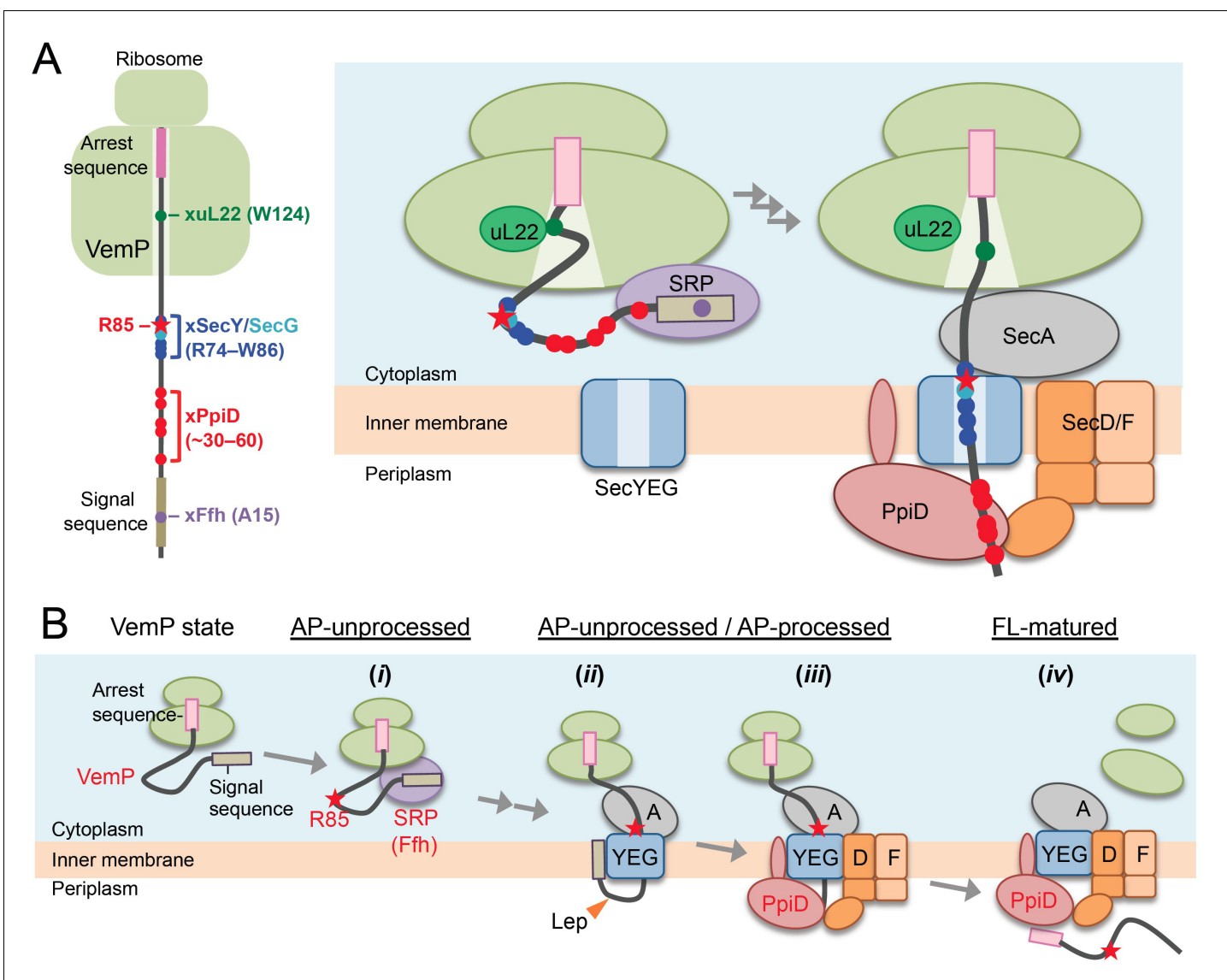

**Figure 6.** A model of the arrest-cancelation of a VemP nascent polypeptide. (**A**) Schematic interaction maps of VemP during its translocation processes. The positions of the crosslinking with other factors and Arg-85 are mapped on the schematic picture of the arrested VemP-ribosome complex (*left*). Hypothetical models of the nascent VemP-ribosome complex on the Sec machinery (*right*). See the text for details. (**B**) An overview of the translocation processes of the arrested VemP including the arrest-cancelation steps. See the text for details.

We suggest that VemP in the ribosomal tunnel changes its conformation from the compacted state to some other state, perhaps a more extended one, as the translocation process proceeds, because the structural determination had used a nascent chain-ribosome complex without any translocation factors (*Su et al., 2017*; *Figure 6A*, *right*). It is conceivable that a pulling force generated by translocation acts to disrupt the VemP secondary structure and, consequently, the molecular interactions required for the elongation arrest.

We propose the following scheme for the targeting and translocation processes of VemP accompanied by the translation-arrest and its cancelation (*Figure 6B*). (*i*) In the cytosol, SRP recognizes and binds to the signal sequence of VemP and targets the nascent chain to the translocon. During this co-translational targeting step, the ongoing translation of VemP is subject to elongation arrest. The use of the SRP pathway, in conjunction with the amino acid sequence features of its signal sequence (*Yap and Bernstein, 2011*) and the early mature part (see below), might ensure the timely release of the translation complex from the translation-arrested state. (*ii*) The signal sequence is then inserted into the lateral gate region while the mature region of VemP is going to span the translocon's polypeptide-conducting channel. This early step of translocation is facilitated by the SecA ATPase (*Figure 2—figure supplement 3*) and accompanied by the signal sequence cleavage. Therefore, the inhibition of SecA with $NaN_3$ causes accumulation of the AP-un form of VemP. (*iii*) SecD/F and PpiD then interact with VemP and cooperate in canceling the translation arrest of VemP while aiding its further translocation. (*iv*) Finally, the full-length, mature VemP polypeptide is exported to the periplasm.

In the arrest cancelation event, PpiD and SecD/F seem to cooperate either sequentially or in concert, and our crosslinking results revealed that they physically interact with each other (*Figure 4E*). PpiD was a member of the periplasmic chaperone network (*Dartigalongue and Raina, 1998*; *Matern et al., 2010*), which directly interacts with both SecY/E/G and ribosome-tethered polypeptides (*Antonoaea et al., 2008*; *Sachelaru et al., 2014*). Although an in vitro study revealed that PpiD stimulates translocation of substrate proteins (*Fürst et al., 2018*), the deletion of the *ppiD* gene caused no detectable defect of outer membrane protein biogenesis (*Justice et al., 2005*). We found that the translocation of MBP was compromised weakly in the Δ*ppiD* strain (*Figure 4A*). At any rate, the exact role of PpiD in protein export remains obscure. We showed here that PpiD physically contacts SecD as well as the AP-pro form of VemP in vivo. Furthermore, our genetic and biochemical analyses indicated that PpiD has a role in the cancelation of the VemP translation arrest (*Figure 4*). Although our experiments failed to detect a VemP-SecD crosslinking, due possibly to the transient/ unstable nature of their interaction, SecD/F also functions in the arrest cancelation of VemP (*Figure 4D*). We propose that PpiD and SecD/F cooperate to facilitate translocation and arrest-cancelation of VemP.

SecD/F works as a monovalent cation-driven motor that pulls up a translocating polypeptide by undergoing a conformational change of the P1 head (*Tsukazaki et al., 2011*). Such an action could generate a pulling force that cancels the VemP elongation arrest. By contrast, PpiD itself would not generate a pulling force against the nascent chain because it is an ATP-independent chaperone (*Matern et al., 2010*). It is conceivable that PpiD captures a VemP polypeptide emerging from the translocon and then hand it over to the P1 head domain, which is known as the substrate-binding site of SecD (*Furukawa et al., 2017*). Possibly, the direct contact between PpiD and SecD facilitates the substrate transfer from PpiD to SecD. It is also possible that the binding of VemP by PpiD helps to prevent the reverse movement of VemP. Since both SecD/F and PpiD are well conserved in enterobacteria including *E. coli* that possess no VemP protein, the co-operative function of these proteins could be utilized to stimulate late step translocation of some un-identified secretory proteins in these organisms. Although further studies including identification of contact sites of PpiD with the P1 head of SecD and the detailed in vitro study of the late step translocation of VemP and other substrates are needed, it is noteworthy that bacterial species appear to be equipped with a dedicated pulling system acting from the extra cytosolic location for protein secretion and translational control.

We identified Arg-85 as a *cis*-element important for the in vivo stability of the translation-arrest of VemP, through its role in the regulation of the translocation coupling of the arrest-cancelation. Arg-85 is highly conserved among the VemP orthologues (*Figure 5—figure supplement 3*), pointing to its importance in the VemP function. This residue does not contribute to the establishment, per se, of the translation-arrested state of VemP, when its secretion is blocked (*Figure 5B*). However, under the export-proficient conditions, Arg-85 somehow acts to retard the translocation of VemP

independently of the translocation arrest of VemP, and consequently increase the PpiD-dependency of the VemP translocation (*Figure 5C*). Our crosslinking results show that a region encompassing the Arg-74 to Trp-86 interval interacts with the translocon's components in the arrested state of VemP (*Figure 6A*), indicating that Arg-85 resides in the SecY/E/G-contacting region of the VemP nascent chain (*Figure 6A*). This spatial arrangement leads us to assume that Arg-85 interacts with the translocon, which inhibits the forward movement of VemP and stabilizes its elongation-arrested state by antagonizing the translocation force that otherwise (when Arg85 is mutated) could lead to a rapid arrest cancelation. Thus, a force strong enough to overcome the arrest must be applied in the case of wild-type VemP to make it competent for regulation.

Our experiments demonstrate that the presence of Arg-85 acts as an obstacle for translocation. However, the trans-side motor SecD/F with the aide from PpiD would overcome the obstacle and manage to allow translocation and arrest-cancelation to take place with a physiological efficiency. When Arg-85 is mutated to other amino acid residues, translocation does not meet any obstacle and proceed very rapidly such that we do not see any significant arrest in the Sec-proficient cells. When the signal sequence of VemP is non-functional or the Sec system is inactivated, the elongation arrest occurs normally and continues stably whether or not Arg85 is mutated.

In any case, the VemP arrest cancelation may require a pulling force by the PpiD-SecD/F system in addition to the force generated by SecA, which alone is insufficient. From this line of considerations, we propose that translation and translocation of VemP are tuned to monitor the periplasmic motor activity of the bacterial Sec system. Remarkably, monitoring substrates employ different force-generation mechanisms to enable real-time monitoring of local events of polypeptide translocation, which is superimposed with temporal elements of translation. The notion that the conserved Arg-85 residue confers the PpiD-SecD/F dependency of the arrest-release of VemP requires further experimental verification, including in vitro recapitulation of arrest cancelation by SecA plus ATP vs PpiD-SecD/F plus proton/sodium-motive forces. A single-molecule analysis to measure the pulling force required for the arrest-release of VemP will also be informative.

Interestingly, SecM also possesses a *cis*-element that is located outside the ribosome in the arrested translation complex and required for the efficient arrest cancelation (*Ito et al., 2018*; *Nakamori et al., 2014*). The regulatory target of SecM is SecA (*Nakatogawa and Ito, 2001*), and its arrest is canceled at an early stage of translocation, before the cleavage of the signal sequence (*Mori et al., 2018*; *Nakatogawa and Ito, 2001*). SecM may monitor the SecA-stimulated early translocation event via its cis-element. Thus, a role of the cis-element in SecM should be different with that of Arg-85 in VemP. Our current results, taken together, reveal that different monitoring substrates monitor different sub-steps of translocation, and a cis-element aids the monitoring function by different mechanisms. Our results suggest that pulling forces applied to the ribosome-nascent-chain complexes can be diverse and differentiated to be sensed and processed precisely by regulatory nascent polypeptides, opening up a new dimension of the force-sensing translation arrest.

## Materials and methods

### Key resources table

| Reagent type (species) or resource | Designation | Source or reference | Identifiers | Additional information |
|---|---|---|---|---|
| Strain (*E. coli*) | MC4100 | This paper | | *Supplementary file 1* |
| Strain (*V. alginolyticus*) | 138–2 | This paper | | *Supplementary file 1* |
| Strain (P1 bacteriophage) | | Laboratory stock | CGSC12133 | |
| Recombinant DNA reagent | Plasmids | This paper | | *Supplementary file 2* |
| Sequence-based reagent | | This paper | PCR primers | *Supplementary file 3* |
| Antibodies | 1st antibodies | This paper | | listed in the below |
| Antibody | Goat Anti-Rabbit IgG (H + L)-HRP Conjugate | Bio-Rad Laboratories | 1706515 | |
| Chemical compound | H-p-Bz-Phe-OH | Bachem | F2800 | |
| Chemical compound | Methionine, L-[$^{35}$S] Translation Grade | American Radiolabeled Chemicals | ARS 01014 | |

*Continued on next page*

*Continued*

| Reagent type (species) or resource | Designation | Source or reference | Identifiers | Additional information |
|---|---|---|---|---|
| Chemical compound | ANTI-FLAG M2 Affinity Gel | Sigma-Aldrich | A2220 | |
| Chemical compound | nProtein A Sepharose 4 Fast Flow | GE Healthcare | 17528004 | |
| Chemical compound | Ni-NTA Agarose | QIAGEN | 30250 | |
| Commercial kit | ECL Western Blotting Detection Reagents | GE Healthcare | RPN2106 | |
| Commercial kit | ECL Prime Western Blotting Detection Reagents | GE Healthcare | RPN2232 | |
| Software, Algorithm | Microsoft Excel | Microsoft | | |
| Software, Algorithm | Bio-imaging Analyzer BAS-1800, BAS-5000 | Fujifilm/GE Healthcare | | |
| Software, Algorithm | Image Qaunt LAS 4000 mini | Fujifilm/GE Healthcare | | |
| Software, Algorithm | Multi Gauge | Fujifilm/GE Healthcare | | |

## Bacterial strains, plasmids, and primers

*Escherichia coli* K12 strains and *Vibrio alginolyticus* VIO5 strains, plasmids, and primers used in this study are listed in *Supplementary files 1*, *2,* and *3*, respectively. Details of the strain and plasmid construction are described in Construction of Mutant Strains and Plasmid Construction, respectively.

## Media and bacterial cultures

*E. coli* cells were grown in L rich medium (10 g/L bacto-tryptone, 5 g/L bacto-yeast extract, 5 g/L NaCl; pH adjusted to 7.2 with NaOH) or M9 synthetic medium (without $CaCl_2$; *Miller, 1972*) supplemented with maltose (final 0.2%), glycerol (final 0.4%), all amino acids (except Met and Cys; final concentration of 20 µg/mL each). 50 µg/mL ampicillin, 20 µg/mL chloramphenicol, 25 µg/mL kanamycin, 25 µg/mL tetracycline, and 50 µg/mL spectinomycin were added as appropriate for growing plasmid-bearing cells and selection of transformants and transductants. *V. alginolyticus* cells were grown in VC-rich medium (5 g/L bacto-tryptone, 5 g/L bacto-yeast extract, 4 g/L $K_2HPO_4$, 30 g/L NaCl, 2 g/L glucose; *Terashima et al., 2010*). For induction with arabinose, we used modified VC medium containing 0.2% arabinose instead of 0.2% glucose. 2.5 µg/mL chloramphenicol was added as appropriate for growing plasmid-bearing cells. Bacterial growth was monitored with Mini photo 518R (660 nm; TAITEC Co., Saitama, Japan).

## Antibodies

ANTI-FLAG M2 Affinity Gel (anti-FLAG antibody for immunoprecipitation) was purchased from Sigma-Aldrich, Co. LLC (St. Louis, MO). For preparation of an antibody against SecD, two oligopeptides (CYKDSGKKDANG and CYGGKRVLLLSI) were synthesized. They were mixed, conjugated with a carrier protein, keyhole limpet hemocyanin, via the Cys residue attached at their N-terminus and used to raise antibodies in rabbits. Anti-SecD IgGs were affinity-purified and used in the experiments. Anti-uL22, anti-Ffh, and anti-PpiD antibodies were gifts from S. Chiba (Kyoto Sangyo University, Kyoto, Japan), C. A. Gross (University of California at San Francisco, San Francisco, CA) and M. Müller (University of Freiburg, Freiburg, Germany), respectively. Anti-VemP, anti-V.SecD1, and anti-V.SecD2 (*Ishii et al., 2015*), antibodies as well as anti-SecG (*Nishiyama et al., 1993*) and anti-MBP (*Baba et al., 1990*) antibodies were described previously.

## Construction of mutant strains

RM3122 (HM1742, Δ*secG::kan*), RM3124 (HM1742, Δ*ppiD::kan*), and HM4790 (HM1742, Δ*secB::kan*) were constructed by transducing Δ*secG::kan* from JW3142 (*Baba et al., 2006*), Δ*ppiD::kan* from JW0431 (*Baba et al., 2006*) and Δ*secB::kan* from JW3584 (*Baba et al., 2006*) to HM1742 (*Mori and Ito, 2006b*), respectively. HM4798 (HM1742, Δ*secB::FRT*) was constructed from HM4790 by removing the *kan* gene using pCP20 (*Cherepanov and Wackernagel, 1995*). RM2831 (HM1742, *ffh-his₁₀*) were constructed by essentially the same procedure as the construction of SPA-tag collection strains

(*Butland et al., 2005*). First, a *his₁₀-tag::kan* fragment having sequences identical to the upstream and downstream regions of the *ffh* termination codon at its 5' and 3' ends, respectively, was PCR-amplified from pRM573 (See the last part of this session) using ffh-his₁₀-f and ffh-his₁₀-r primers. Then, this fragment was integrated into the *E. coli* DY330 chromosome using the λ-Red recombination system (*Yu et al., 2000*). After transferring an *ffh-his₁₀-tag::kan* to HM1742 by P1 transduction, the *kan* cassette was removed using pCP20 to yield RM2831. RM2834 (HM1742, *secY-his₁₀*) was constructed in a similar way, except that a different pair of the primers, secY-his₁₀-f and secY-his₁₀-r were used. RM2935 (HM1742, *yidC-his₁₀* kan) and RM3032 (HM1742, *ppiD-his₁₀* kan) were also constructed similarly using the primer pairs, yidC-his₁₀-f/yidC-his₁₀-r and ppiD-his₁₀-f/ppiD-his₁₀-r, respectively, whereas the *kan* cassettes of these strains were not removed. AD96 was constructed by transducing *secA51*(Ts) from MM52 (*Oliver and Beckwith, 1981*) to KI269 (*Akiyama and Ito, 1985*).

*Vibrio* mutant strains RMV2 and RMV7 were constructed using a 'suicide vector', pSW7848, carrying the toxin-encoding *ccdB* gene under the arabinose promoter control (*Ishii et al., 2015*). A total of 150 μL of an overnight culture of *E. coli* β3914 cells harboring pRM691 or pRM744 was mixed well with 50 μL of an overnight culture of *V. alginolyticus* VIO5 strain. Cells were harvested and resuspended in 100 μL of the VC medium and 2.5 μL of the suspensions was spotted on the VC agar medium containing 300 μM 2,6-diaminopimelic acid (DAP) and incubated at 30°C for 6 hr. Then, the cells were streaked on a VC plate containing 2.5 μg/mL chloramphenicol but without DAP to select *Vibrio* cells in which a plasmid had been integrated on the chromosome. Subsequently, the chloramphenicol-resistant bacteria were grown on VC-0.2% arabinose agar plates to counter-select the plasmid-integrated *Vibrio* cells. After confirmation of chloramphenicol-sensitivity and arabinose resistance of obtained cells, presence of the introduced mutations was confirmed by colony PCR.

## Construction of plasmids

pTS48 was constructed in the same way as the construction of pTS47 (pHM1021-*vemP-3xflag-myc*; *Mori et al., 2018*). pRM374 (pTV118N-*vemP-3xflag-myc*) was constructed by subcloning a NcoI-HindIII fragment carrying the *vemP-3xflag-myc* prepared from pTS48 into the same sites of pTV118N. pHM1021-*vemP(amb)−3xflag-myc* plasmids and pTV118N-*vemP(amb)−3xflag-myc* plasmids shown in *Supplementary file 2* were constructed as follows. An *amber* mutation at the codons corresponding to the amino acid residues, F11, M16, A21, F24, K31, Y36, Q41, S46, N51, F56, E61, S66, S71, D76, F81, W86, R91, D96, V101, N106, V111, D121, Q126, F131, or S136 was introduced into pTS48 by site-directed mutagenesis using a pair of appropriate primers. Then, pTV118N-*vemP-3xflag-myc* plasmids containing the same *amber* mutation was constructed by cloning the NcoI-HindIII fragments that had been prepared from the resultant pHM1021-based plasmids into the same sites of pTV118N. For construction of plasmids carrying an *amber* codon at positions of the other amino acid residues, an *amber* mutation was first introduced into pRM374 by site-directed mutagenesis. For construction of pHM1021-*vemP-3xflag-myc* plasmids containing the same *amber* mutation, the NcoI-HindIII fragments were prepared from the pRM374-based plasmids and sub-cloned into the same sites of pHM1021. Derivatives of pTS48 encoding a VemP-F₃M mutant with an amino acid alteration were constructed by site-directed mutagenesis. pRM848 (pHM1021-*vemP(F.S., R85W)−3xflag-myc*) was constructed from pHM1202 (pHM1021-*vemP(F.S.)−3xflag-myc*) by site-directed mutagenesis. pRM662 (pBAD24-*vemP(R85W)-V.secD2/F2*) was constructed from pHM810 (pBAD24-*vemP-V.secD2/F2*; *Ishii et al., 2015*) by site-directed mutagenesis. The pHM1021-*vemP-secDF2_{VA}* (pRM663), pHM1021-*vemP(W143A)-V.secD2/F2* (pRM666), and pHM1021-*vemP(R85W)-V.secD2/F2* (pRM667) plasmids were constructed by subcloning the NcoI-SphI fragment carrying the *V.vemP-secD2/F2* genes with or without the respective mutations that had been prepared from pBAD24-*V.vemP-secD2/F2* plasmids (pHM810, pHM846 and pRM662, respectively). pRM520 (pTV118N-*his₁₀-vemP-3xflag-myc*) was constructed as follows. A *his₁₀-vemP-3xflag-myc* fragment was PCR-amplified from pTS48 using his₁₀-vemP-f and M4C primers, digested with NcoI and HindIII, and cloned into the same sites of pTV118N. pRM557 (pTV118N-*his₁₀-vemP(ΔSS)−3xflag-myc*) was constructed by site-directed mutagenesis of pRM520. Derivatives of pRM557 (pRM562 and pRM563) containing a mutation in the *vemP* gene were also constructed by site-directed mutagenesis.

pRM83c was a constitutive expression vector, in which the operator region *lacO1* on the *lac* promoter of pRM83 (*Miyazaki et al., 2016*) had been converted from AATTGTGAGCGGATAACAATT to AATT<u>A</u>T<u>TGTTA</u>GA<u>C</u>A<u>A</u>T<u>A</u>ATT (the mutated residues are underlined) by successive site-directed mutagenesis as follows. First, the operator region *lacO1* of pRM83 was changed from AATTG

TGAGCGGATAACAATT to AATT**ATTGT**CGGATAACAATT using lacO1-c-f and lacO1-c-r primers (the mutations introduced were shown in bold). The second mutagenesis was done with the resultant plasmid as a template using lacO1-c2-f and lacO1-c2-r primers to introduce two additional mutations (AATTATTGTCGGACAATAATT). Finally, further site-directed mutagenesis (AATTATTGTTAGACAA TAATT) was conducted using lacO1-c3-f and lacO1-c3-r primers to obtain pRM83c. pRM656 (pRM83c-*his₁₀-secD/secF*) was constructed as follows. A *his₁₀-secD/secF* fragment was PCR-amplified from pHM735 (*Tsukazaki et al., 2011*) using his₁₀-secD-f and secF-r primers, digested with HindIII and SalI, and cloned into the same sites of pRM83c. To reduce the expression level of SecD/SecF, the start codon (ATG) for *secD* on the above plasmid was mutated to TTG by site-directed mutagenesis. pRM661 (pRM83c-*ppiD*) was constructed as follows. A *ppiD* fragment was PCR-amplified from a *ppiD-his₁₀* plasmid (laboratory stock) using ppiD-f and ppiD-r primers, digested with HindIII and SalI, and cloned into the same site of pRM83c.

The plasmid used for construction of RMV2 (VIO5, Δ*ppiD*) was constructed as follow. First, a DNA fragment containing a *V.ppiD* gene with an (1 kb) upstream and downstream sequences was amplified from the genome of VIO5 using Va-ppiD-f and Va-ppiD-r primers, and ligated with the BamHI- and SalI-digested pUC118 fragment using In-Fusion HD cloning Kit to generate pRM670. Next, a DNA fragment for the *V.ppiD*-upstream and downstream region with the entire vector sequence was amplified from pRM670 by PCR using del-Va-ppiD-f and del-Va-ppiD-r primers, and self-ligated using In-Fusion HD cloning Kit to produce pRM674. Finally, ~2 kbp DNA fragment containing *V. ppiD*-upstream and -downstream regions was amplified from a pRM674 using pSW-ppiD-f and pSW-ppiD-r primers, and ligated with the NaeI-digested pSW7848 using In-Fusion HD cloning Kit to construct pRM691. The plasmid used for construction of RMV4 (VIO5, P*ₐᵣₐ-V.ffh*) was constructed as follow. First, a DNA fragment containing an *V.ffh* gene was amplified from the genome of VIO5 using Va-ffh-f and Va-ffh-r primers, and ligated with the NcoI- and SphI-digested pBAD24 fragment using In-Fusion HD cloning Kit to generate pRM737. Next, an *V.ffh*-upstream region that had been amplified from the VIO5 genome using u-ffh-f and u-ffh-r primers and a DNA fragment that had been amplified from the pRM737 using pRM737-f and pRM737-r were ligated using In-Fusion HD cloning Kit to generate pRM740. Finally, ~3.8 kbp DNA fragment was amplified from the pRM740 plasmid using pSW-ffh-f and pSW-ffh-r primers, and ligated with the NaeI-digested pSW7848 using In-Fusion HD cloning Kit to construct pRM744.

pRM570 (a plasmid carrying the *spa-tag* and a *kan* cassette *sequences* (*spa-tag::kan*)) was constructed as follows. A *spa-tag::kan* fragment was amplified from the genome of an *ffh-spa-tag* strain (*Butland et al., 2005*) using spa-kan-f and spa-kan-r primers, digested with EcoRI and SalI, and cloned into the same sites of pUC118. The stop codon of *spa-tag* was changed from TAG to TAA by site-directed mutagenesis to generate pRM570. To construct pRM573 (pUC118-*his₁₀-tag::kan*), a *his₁₀-tag::kan* fragment was amplified from pRM570 using *his₁₀*-kan-f and M4C primers, digested with EcoRI and SalI, and cloned into the same sites of pUC118.

## Immunoblotting analysis

This method was used in *Figures 3A–C* and *4C–E* and *Figure 5—figure supplement 1D*. Solubilized total proteins were separated by SDS-PAGE and electro-blotted onto a PVDF membrane (Merck Millipore; Billerica, MA). The membrane was first blocked with 5% skim milk in PBST (Phosphate Buffered Saline with Tween 20), and then incubated with anti-SecD (1/2,000 dilution), anti-V.SecD1 (1/2,000), anti-V.SecD2 (1/2,000), anti-PpiD (1/20,000 or 1/50,000), anti-Ffh (1/10,000) or anti-VemP (1/2,000) antibodies After washing with PBST, the membrane was incubated with a horseradish peroxidase (HRP)-conjugated secondary antibody (1/5,000) (Goat Anti-Rabbit IgG (H + L)-HRP Conjugate; Bio-Rad Laboratories, Inc, Hercules, CA) in PBST. Proteins were visualized with ECL Western Blotting Detection Reagents (GE Healthcare UK Ltd, Amersham Place Little Chalfont, England) or ECL Prime Western Blotting Detection Reagents (GE Healthcare) and LAS4000 mini lumino-image analyzer (GE Healthcare).

## PiXie analysis

This analysis was used in *Figures 1B–F* and *2A*, *Figure 1—figure supplements 1A* and *2–5* and *Figure 2—figure supplement 1*. Cells were first grown at 30℃ in M9-medium supplemented with 2 μg/mL thiamine, 0.4% glycerol, 0.2% maltose, all amino acids (except Met and Cys; final concentration

of 20 µg/mL each), 0.5 mM pBPA (H-p-Bz-Phe-OH F-2800; Bachem AG, Bubendorf, Switzerland), and 0.02% arabinose until early log phase. After IPTG induction for 15 min, cells were pulse-labeled with 370 kBq/mL [35S]Met (American Radiolabeled Chemicals, Inc, St. Louis, MO) for 30 s. In the case of the experiments in *Figure 2—figure supplement 4*, 0.02% NaN3 and [35S]Met were added simultaneously. After addition of excess non-radioactive Met (final conc. 250 µg/mL), a 350 µl portion of the cell cultures was quickly removed and put into wells of a 24-well microtiter plate (AGC Tehcno Glass Co. Ltd, Shizuoka, Japan) on a temperature-controllable and movable stage (MS Tech Co. Ltd., Kyoto, Japan). Cells were kept at 30°C and UV-irradiated for 1 s at appropriate time points during chase using SP-9 equipped with an SFH lens (USHIO Inc, Tokyo, Japan) at a distance of 5 cm. Control samples without UV-irradiation in *Figure 2A* and *Figure 2—figure supplement 1* were also put to the microtiter plate on the stage and treated in the same way except that they were not UV-irradiated. After UV irradiation, total cellular proteins were immediately precipitated with 5% trichloroacetic acid (TCA), washed with acetone, and solubilized in SDS-buffer (50 mM Tris-HCl (pH 8.1), 1% SDS, 1 mM EDTA). The samples were then diluted 33-fold with Triton-buffer (50 mM Tris-HCl (pH 8.1), 150 mM NaCl, 2% Triton X-100, 0.1 mM EDTA). After clarification, samples were incubated with appropriate antibodies and nProtein A Sepharose 4 Fast Flow (GE healthcare) or Ni-NTA Agarose (QIAGEN) at 4°C over-night with slow rotation. Proteins bound to the antibody/ProteinA-Sepharose or Ni-NTA Agarose were recovered by centrifugation, washed with Triton buffer and then with 10 mM Tris-HCl (pH 8.1) and eluted by incubation at 37°C for more than 5 min in SDS-sample buffer (62.5 mM Tris-HCl (pH 6.8), 2% SDS, 10% glycerol, 5 mg/mL bromophenol blue). The isolated proteins were separated by SDS-PAGE, and visualized with BAS1800 or BAS5000 phosphoimager (Fujifilm Co., Tokyo, Japan). Band intensities were quantified using MultiGauge software (Fujifilm). To obtain the graphs in *Figure 2B*, the band intensity of VemP-FL (open diamonds), VemP-APs (AP-un + AP-pro; the signal intensity of the AP-pro is normalized by the Met content) (closed diamonds) and XLs (colored symbols) in *Figure 2A* was quantitated.

## In vivo stability of the arrest-form of VemP

The procedure was used in *Figures 3A*, *4A, D* and *5*, *Figure 2—figure supplement 3*, *Figure 3—figure supplement 1* and *Figure 5—figure supplements 1B* and *2*. Cells were first grown at 30°C in M9-medium supplemented with 2 µg/mL thiamine, 0.4% glycerol, 0.2% maltose, all amino acids (except Met and Cys) with or without 0.05% arabinose until early log phase. After induction with for 15 min, cells were pulse-labeled with 370 kBq/mL [35S]Met for 30 s. In the case of *Figure 2—figure supplement 3* (for analysis of secA51(Ts)), the induction of VemP-F3M was done, at 2 hr after the temperature shift to 42°C. At appropriate time points after addition of excess nonradioactive Met (final conc. 250 µg/mL), total cellular proteins were precipitated with 5% TCA, washed with acetone, and solubilized in SDS-buffer. The samples were subjected to the IP as described above. Isolated proteins were separated by SDS-PAGE, and visualized with BAS1800 phosphoimager. Percentages of the arrested VemP were calculated by the following equation: arrested VemP (%)=100 × [VemP-APs]/[(VemP-FL) + (VemP-APs)], where VemP-APs (in *Figure 2B*) and VemP-FL are the intensities of the respective bands.

## In vivo photo-crosslinking analysis

In the *Figure 4E*, Cells were grown at 37°C in L medium containing 0.5 mM pBPA until early log phase and induced with 0.02% arabinose for 1 hr. The half volume of the cell cultures was put on a petri dish at 4°C and UV-irradiated for 10 min using B-100AP UV lamp (365 nm; UVP, LLC., Upland, CA), at a distance of 4 cm. The other half was kept on ice as non-UV-irradiated samples. Total cellular proteins were precipitated with 5% TCAs, washed with acetone, and suspended in SDS-sample buffer. The samples were subjected to SDS-PAGE and immunoblotting analysis.

## Acknowledgements

We thank Shinobu Chiba, Carol A Gross and Matthias Müller for providing antibodies, NBRP (NIG, Japan): *E. coli* for bacterial strains, Koreaki Ito for critical reading and editing of the manuscript, and our laboratory members for discussion.

## Additional information

### Funding

| Funder | Grant reference number | Author |
|---|---|---|
| Japan Society for the Promotion of Science | 18H06047 | Ryoji Miyazaki |
| Japan Society for the Promotion of Science | 19K21179 | Ryoji Miyazaki |
| Japan Society for the Promotion of Science | 20K15715 | Ryoji Miyazaki |
| Japan Society for the Promotion of Science | 15H01532 | Yoshinori Akiyama |
| Japan Society for the Promotion of Science | 18H02404 | Yoshinori Akiyama |
| Japan Society for the Promotion of Science | 17H05666 | Hiroyuki Mori |
| Japan Society for the Promotion of Science | 17H05879 | Hiroyuki Mori |
| Japan Society for the Promotion of Science | 17K07334 | Hiroyuki Mori |
| Japan Society for the Promotion of Science | 20K06556 | Hiroyuki Mori |

The funders had no role in study design, data collection and interpretation, or the decision to submit the work for publication.

### Author contributions

Ryoji Miyazaki, Conceptualization, Resources, Data curation, Funding acquisition, Validation, Investigation, Methodology, Writing - original draft; Yoshinori Akiyama, Resources, Supervision, Funding acquisition, Validation, Writing - review and editing; Hiroyuki Mori, Conceptualization, Resources, Supervision, Funding acquisition, Validation, Investigation, Methodology, Writing - original draft, Writing - review and editing

### Author ORCIDs

Yoshinori Akiyama http://orcid.org/0000-0003-4483-5408
Hiroyuki Mori https://orcid.org/0000-0002-0429-1269

### Decision letter and Author response

Decision letter https://doi.org/10.7554/eLife.62623.sa1
Author response https://doi.org/10.7554/eLife.62623.sa2

## Additional files

### Supplementary files

- Supplementary file 1. Table S1. Strains used in this study

- Supplementary file 2. Table S2. Plasmids used in this study

- Supplementary file 3. Table S3. Primers used in this study

- Transparent reporting form

### Data availability

All data generated and analyzed during this study are included in the manuscript and supporting files. Source data files have been provided for Figures 2, 3 and 4, Figure 2—figure supplements 1, 3 and 4 and Figure 5—figure supplements 1 and 2.

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

## Appendix 1

## Interactions of a VemP nascent chain with Sec translocon and the related components

As described in the main text, the interaction of the signal sequence of VemP with SRP occurs during the targeting of VemP to the translocon. According to the widely accepted model for co-translational translocation, the signal sequence of VemP should interact with the lateral gate of SecY channel and be intercalated into the gate to move towards a lipid phase, at the initial stage of the translocation, following the targeting process (*Van den Berg et al., 2004*). It is expected that during the above 'partition' process, *p*BPA introduced into the signal sequence might be cross-linked with the lateral gate of SecY. However, we have not detected such SecY-VemP XLs. The signal sequence of VemP is mostly removed at an early stage of translocation, where translation of VemP still remains arrested on the translocon (note that even after pulse-labeling, the arrested VemP molecules are mostly in the processed (AP-pro) form). Thus, even if the *p*BPA-containing signal sequence of VemP, which had been cleaved off form AP-un, is crosslinked with the SecY, the XLs cannot be isolated by immunoprecipitation using the anti-VemP antibody raised against the epitope in the mature part of VemP (*Ishii et al., 2015*). Although the XLs might be able to be isolated by using a His-tag attached to SecY, a small size of the signal sequence will cause only a slight retardation in the mobility of the XL bands, making it difficult to distinguish them from a broad SecY band. Also, a low amount of AP-un having *p*BPA in the signal sequence will hinder detection of its crosslinking with SecY.

We found that the residue 30–60 region and the residue 74–86 region of VemP were corsslinked with PpiD and SecY, respectively (*Figure 6B*). These results can be nicely explained if we assume that the arrested VemP is inserted into translocon as a hairpin-like structure with its signal sequence partitioned into the lateral gate, as, in this case, the former region will be close to PpiD in the periplasm whereas the latter region will be within or near the translocon channel. If this assumption is correct, the residue 81 and 83 of VemP that were crosslinked with SecG are expected to be located near the cytoplasmic cavity of the SecY channel. The crystal structure of the SecYEG complex determined by *Tanaka et al., 2015* reveals that a cytoplasmic loop between the first and the second TM segments of SecG covers the cytoplasmic cavity of the SecY channel at a resting state. It would be thus conceivable that even at the active state, the cytoplasmic loop of SecG exists near the cytoplasmic cavity of SecY. This would provide one explanation for why SecG as well as SecY is crosslinked with a translocating VemP.

Since several studies (*Petriman et al., 2018*; *Sachelaru et al., 2013*) reported that the Sec translocon directly interacts with the membrane chaperone, YidC having a large periplasmic domain, one would expect that VemP can be cross-linked with YidC as well. However, we detected no YidC-VemP XL at least in this study. Recent papers suggest that PpiD and YidC associate with Sec translocon to form two distinct subassemblies (*Götzke et al., 2014*; *Jauss et al., 2019*). No crosslinking of VemP with YidC can be explained if a nascent VemP polypeptide can specifically recognize and utilize the Sec translocon that forms the complex with PpiD/SecDF to effectively monitor SecD/F function.

