## [Decision Letter]

**Acceptance summary:**

The study investigates the *Vibrio* "secretion monitor" VemP, that monirots the secretion capacity of an organism using ribosome stalling. This study has discovered that translation stalling of the secretion monitor VemP in V. alginolyticus is relieved (or cancelled) by the action of PpiD, a periplasmic chaperone that acts with the proteins SecD/F. Failure to relieve the stall allows increased expression of SecD/F to complete this auto-regulatory circuit. The authors further identify a crucial conserved amino acid (R85) in VemP that imposes its dependence on PpiD for translocation.

**Decision letter after peer review:**

Thank you for submitting your article "Fine interaction profiling of VemP and mechanisms responsible for its translocation-coupled arrest-cancelation" for consideration by *eLife*. Your article has been reviewed by Suzanne Pfeffer as the Senior Editor, a Reviewing Editor, and three reviewers. The reviewers have opted to remain anonymous.

The reviewers have discussed the reviews with one another and the Reviewing Editor has drafted this decision to help you prepare a revised submission.

The study investigates the *Vibrio* "secretion monitor" VemP, a less well understood analog of the better studied SecM of *E. coli*. Although VemP and SecM are not homologous, they are conceptually similar in serving the function of monitoring the secretion capacity of an organism using ribosome stalling. In the SecM system, a translation stall that occurs early during synthesis is relieved when SecA "pushes" the nascent chain during translocation and thereby applies a stall-cancelling force. When SecA is limiting, the stall persists, which allows SecA to be translated as it is located in the same operon as SecM. The VemP system operates in a roughly analogous way, but exactly how its stall is relieved (i.e., what activity is being monitored by this system) is not as well understood. As VemP is in the SecD/F operon, presumably this is involved, but has not been shown.

To investigate this problem, the authors use a site-specific crosslinking approach that has good time resolution. By systematically placing the photo-crosslinking amino acid along the length of VemP, they nicely map all the physical interactions. Using molecular weight, IPs, and analysis in deletion strains, most (but not all) of the main crosslinking partners are identified. This analysis paints a picture of a series of interactions that involve the ribosome exit tunnel, Ffh (the bacterial SRP homolog), the SecYEG translocon, and the periplasmic protein PpiD. Genetic analysis provides support that Ffh is involved in targeting (an expected result), and that PpiD and SecD/F are involved in cancelling the VemP stall. Crosslinking suggests that PpiD directly interacts with SecD/F, suggesting that they cooperate in stall cancellation. An interesting side observation is that mutation of a conserved arginine in VemP (R85), that seems to be within the SecY channel in the stalled state, causes efficient stall cancellation.

The strengths of the study are the interesting question that is being addressed, the high technical quality of the data, and the very systematic analysis of physical interactions using their time-resolved crosslinking method (termed PiXie). The conclusion that SecD/F and PpiD are involved in cancelling the stall at a late stage of translocation is generally convincing. The study would benefit from some greater specificity controls in a couple of key areas. One item that is poorly examined is the role of R85. Although an extensive analysis of this intriguing finding is beyond the scope of the study, some very straightforward characterisation would substantially add to the study and should be considered.

Essential revisions:

1) The authors need to test the relationship between R85 and the dependence for translocation on PpiD and/or SecDF. The reason this is the most important is because these are the newest and most interesting aspects of the study: R85 is the first example of a residue outside the ribosome tunnel that influences stalling/restarting, and this is the first paper to demonstrate that SecD/F and PpiD are involved in stall cancellation. Whether R85 is important because of this residue imposing PpiD or SecD/F dependence seemed to me essential to address.

The most interesting and satisfying result would be if R85 imposes dependence on SecD/F and PpiD. This would mean that VemP has evolved this otherwise unnecessary residue in order to sense functionality of these factors so it can respond accordingly when they are impaired.

2) Although the authors stated that "the results shown are representatives of three technical replicates" in the legend to Figure 2A, the reviewers ask the authors to show other data using biological replicates to confirm the reproducibility in a supplemental figure. If there is an interactor, the kinetics of which is not reproducible, the reviewers recommend omitting the interactor. Such omission would not change the essential point in this work. Also, overall data interpretation by the authors is not quantitative and thus very difficult to follow. In the revised manuscript, the authors should disclose how they assigned the bands and quantitated the intensities in more detail, not only for Figure 2 but also for other figures, in particular Figure 1.

3) In addition, proof that SecA is involved as their model suggests would complete the story. This should also be a straightforward experiment considering the expertise of the authors.

Reviewer 1:

1) All the VemP-interactors has been identified from the anticipation based on the molecular weights of the crosslinked products. Since the crosslinked products are so many other than those the authors selected, there would be other interactors that might be involved in the arrest/cancelation process. Please discuss.

2) Is the proposed mechanism to pull the VemP nascent chain applicable to that for SecM?

3) The deletion of the *ppiD* gene causes no detectable defect of outer membrane biogenesis (Justice et al., 2005). Is the contribution of PpiD on the translocation specific to VemP?

4) Some of the labels in Figures (e.g., Figure 1—figure supplement 4, gel for secY-his10 strain) are incorrect.

Reviewer 2:

1) The authors find that Ffh, SecD/F, and PpiD all impact stall cancellation. The specificity of these observations would be greater if the authors could show that a translocon component that is not involved in VemP translocation does not impact the stall or cancellation. For example, does acute depletion of YidC have any effect? This experiment, if feasible, is important to exclude the possibility that any perturbation to protein translocation would impact VemP.

2) The crosslinking result in Figure 4E showing the SecD-PpiD interaction would be more convincing and specific if the BpA were placed in another position in the periplasmic side of SecD/F and shown to NOT crosslink to PpiD. This would help exclude the possibility of promiscuous crosslinking from any periplasmic region of SecD/F due to its general proximity to the periplasmic chaperone PpiD.

3) The authors propose that SecA is involved in translocation based on the effect of N3. However, they do not document an interaction of substrate with SecA or a genetic requirement. A substrate interaction would seem to be a strong prediction of their model. As a number of unidentified high molecular weight crosslinking products are observed in Figure 1—figure supplement 1A, it seems plausible that a direct interaction with SecA is indeed detectable. Have the authors checked this? If so, this would strengthen their study because all physical interactions show in their model would have experimental support (and it would really demonstrate how their PiXiE method can be comprehensive).

4) The discovery of a specific role for R85 is the most intriguing finding in the study, but is hardly explored at all. While many experiments could be proposed that are beyond the scope of this study, the one experiment that I feel is within the scope is to document the (potential) relationship between R85 and translocation-dependence on SecD/F/PpiD. If I have understood the author's model, R85 partially 'clogs' in the SecYEG translocon, which makes translocation dependent on SecD/F/PpiD. It is this dependence on SecD/F/PpiD that makes VemP a monitor of a late step in translocation. It seems important to test whether R85 makes VemP translocation dependent on PpiD, independent of arrest. In this experiment, one could analyse W143A versus W143A/R85A for translocation in WT versus ∆PpiD strains. The aim here is to test whether R85 imposes PpiD-dependence to translocation (by monitoring signal cleavage). If so, this would indicate that VemP has evolved and retained R85 in order to make its translocation and arrest cancellation dependent on PpiD, as speculated by the authors.

5) The colour schemes in Figure 1A and Figure 1—figure supplement 1B are not concordant (for example, uL22 is green in Figure 1A, but in Figure 1—figure supplement 1B green is a 100 kD crosslink; same for other partners). It would be more helpful to a reader if these colours matched up. Perhaps a good solution is to make a hybrid figure that combines all of these observations into one nice panel that is provided in the main figure as a revised Figure 1A. This way, a reader can appreciate an overview of all the crosslinks observed (i.e., Figure 1—figure supplement 1B) together with the subset that were verified by IPs (i.e., Figure 1A). One way to do this is to simply take Figure S1B, and under the respective sized crosslinks, put the corresponding protein and dots where this was verified by IPs (i.e., the information Figure 1A).

6) In the Discussion, the authors claim that their uL22 crosslink is in vivo validation of specific interactions seen by cryo-EM. This conclusion is probably overstated. The photo-cross-linking documents close proximity, but cannot be used to infer specific residue contacts. Perhaps reword this to say that the crosslinking "are consistent with" the previous structure.

Reviewer 3:

1) Most of the experiments were performed by expressing *V. alginolyticus* VemP in *E. coli* as heterologous host. I assume that this is necessary because an orthogonal tRNA/tRNA synthetase system for pBpa insertion does not exist for *V. alginolyticus*. This should be mentioned. It could also help if the authors would briefly summarize what is known about SecYEG, SecA, SRP etc in *V. alginolyticus*. This could strengthen their arguments that the data they observe in *E. coli* indeed reflect what likely is happening in the native host.

2) Ffh was found to cross-link to only one position within the signal sequence (A15), which I found surprising (Figure 1E), considering that approx. 15 residues of the signal sequence were tested (Figure 1—figure supplement 1). Residue A15 and H22 also show cross-linking products of 150 kDa and > 150 kDa, what are those? The predicted Ffh cross-link is also very weak and not detected by the VemP antibodies. Only after pull down with Ni-NTA reacting with His-tagged Ffh, the cross-link is visible. The predicted Ffh-VemP cross-link migrates at 55-60 kDa, which does not seem to fit the expected size (50 kDa + 15-17 kDa)? Are the authors really sure that this reflects an SRP cross-link?

3) It would have been expected that the authors also observe XL to SecA, which based on their model should interact with the mature domain of VemP. Has this been tested and have the high MW XL at 150 kDa and > 150 kDa (Figure 1—figure supplement 1) been probed with anti-secA antibodies?

4) Considering their model, it would have been expected that SecY cross-links also to the signal sequence, if this is trapped at the lateral gate during translocation. Trapping of the signal sequence and a hairpin-like insertion of the nascent VemP would explain the cross-links to PpiD at residues 30-60 and cross-links to SecY of down-stream residues. This could suggest that these down-stream residues are inside the SecY channel. Then why do we see SecG? How does this fit to the general geometry of the SecYEG translocon?

5) The sequential interaction profile of VemP is very interesting, although I think one has to be very careful with the interpretation: there is a substantial variation in the amount of AP-un, AP-pro and FL VemP for the different pBpa variants displayed in Figure 2. Therefore, the rather small differences in the decrease of XL products could simply reflect these variations.

Thus, although their interpretation is probably correct and fits to the general concept of co-translational targeting, the quantification shown in Figure 2 likely overestimates the accuracy of the cross-linking method.

6) I understand that the authors found no XL to YidC, although it is in close contact to SecY and has like PpiD a large periplasmic domain. The authors might want to refer to the papers of Götzke et al., (2014) and Jauss et al., (2019), who suggested that PpiD and YidC associate with distinct SecYEG populations. This could explain why PpiD but not YidC is found in their cross-linking approach.

7) The authors should explain why PpiD shows two cross-linking products. Does this reflect XL to AP-un and AP-pro? The same applies to Figure 4E: why does SecD359 shows two PpiD cross-linking products of significant size difference (approx. 25 kDa)?

8) The use of the secY25/pSTV28-syd strain as translocation defective strain is probably not the best choice. Why not using a more defined SecY mutant, like the SecY39 or SecY40 strains that have been characterized in Koreaki Itos lab and other labs. The partial suppression of the secY24 mutant by overexpression of the largely uncharacterized protein Syd introduces an ambiguity that could be easily prevented. Along the same line: if the secY25/pSTV28-syd is indeed translocation deficient, isn't it surprising that full-length VemP and processed VemP/MBP are detectable at all?

9) A general comment regarding the figures and XLs: the color of the arrow heads is difficult to see (with the exception of red) and the color code in the main text and Figure 1—figure supplement 1B does not match.

---

## [Author Response]

Essential revisions:1) The authors need to test the relationship between R85 and the dependence for translocation on PpiD and/or SecDF. The reason this is the most important is because these are the newest and most interesting aspects of the study: R85 is the first example of a residue outside the ribosome tunnel that influences stalling/restarting, and this is the first paper to demonstrate that SecD/F and PpiD are involved in stall cancellation. Whether R85 is important because of this residue imposing PpiD or SecD/F dependence seemed to me essential to address.The most interesting and satisfying result would be if R85 imposes dependence on SecD/F and PpiD. This would mean that VemP has evolved this otherwise unnecessary residue in order to sense functionality of these factors so it can respond accordingly when they are impaired.

We greatly appreciate this highly helpful advice for increasing significance of our finding. According to the suggestion (comment #4) form the reviewer 2, we addressed whether the mutation of the R85 residue cancels the dependency of the VemP translocation on PpiD in the absence of the translation arrest. We first examined effects of the R85W and the W143A mutations, individually or in combination, on the VemP translocation in the wild type and the Δ*ppiD* strains (the data is shown as Figure 5—figure supplement 2). Although the data apparently support that Arg-85 imposes PpiD-dependence in translocation of VemP, it was difficult to obtain a clear conclusion from the data, because the VemP(W143A) underwent reduced, but still significant elongation arrest in the Δ*ppiD* strain. Thus, we next used a VemP(F.S.) mutant, in which the VemP arrest motif sequence had been replaced with an completely different amino acid sequence by a frame shift (F.S.) mutation (Mori et al., (2018)), which allowed us to examine the effect of the R85W mutation on PpiD dependent-VemP translocation under the condition where contribution of the elongation arrest is completely eliminated. The results clearly showed that Arg-85 confers PpiD-dependence to VemP (the experimental data are presented as new Figure 5C). We have described these results and modified the Discussion section accordingly in the revised manuscript. In addition, we have mentioned these results in the Abstract as they should be one of the most important findings obtained in this study, as suggested by the reviewers.

Please note that we have removed the original Figure 5B that represents quantified results of the original Figure 5A, because the data shown in the original Figure 5A was quite clear and the graph in the Figure 5B seemed to be not so informative.

2) Although the authors stated that "the results shown are representatives of three technical replicates" in the legend to Figure 2A, the reviewers ask the authors to show other data using biological replicates to confirm the reproducibility in a supplemental figure. If there is an interactor, the kinetics of which is not reproducible, the reviewers recommend omitting the interactor. Such omission would not change the essential point in this work.

Thank you for the suggestion. We repeated the PiXie experiments for all the VemP(*p*BPA) derivatives used in the original Figure 2 and Figure 2—figure supplement 1 by using 3 different transformants (biological replicates) to confirm the reproducibility. Although we obtained essentially the same results as the original ones for all the samples examined, the SecY-VemP XLs migrated somewhat broadly on SDS-PAGE and were overlapped with background bands. Unfortunately, in these experiments, the intensities of the background signals substantially were high with unknown reasons. This made exact quantification of the SecY-VemP XLs and their kinetic analysis difficult because they overlap with some of rather strong background signals. Thus, according to the reviewing editor's suggestion, we omitted the kinetic analysis data for the SecY-VemP XLs. Finally, we have removed the original Figure 2 and a part of the original Figure 2—figure supplement 1, and instead included the new Figure 2, in which all the new results of the PiXie experiments obtained with 6 kinds of VemP(*p*BPA) derivatives are presented.

Also, overall data interpretation by the authors is not quantitative and thus very difficult to follow. In the revised manuscript, the authors should disclose how they assigned the bands and quantitated the intensities in more detail, not only for Figure 2 but also for other figures, in particular Figure 1.

In the experiments shown in the new Figure 1 and the new Figure 1 related supplementary figures, we first assigned bands as the XLs when they were specifically pulled-down by the anti-VemP antibody in the VemP(pBPA)-dependent manner. Regarding the VemP-Ffh XL, we could not detect XLs by using the anti-VemP antibody because a strong background band migrated at the overlapped position with the XL. We thus detected the XL as the one that was specifically pulled-down by Ni-NTA agarose in the VemP(A15pBPA)-dependent manner, when cells expressing the Ffh-His_10_ was used (Figure 1E, right panel). The assignments were confirmed by one or some of the following experimental results, (i) reactivity of the XLs to the antibody against partner proteins (uL22 (Figure 1B), SecG (Figure 1C), and PpiD (Figure 1—figure supplement 3B)), (ii) disappearance of the XLs in strains lacking the gene for the partner protein (SecG (Figure 1—figure supplement 3A) and PpiD (Figure 1—figure supplement 3B)), and (iii) pull-down of the XLs from cells expressing His_10_-tagged versions of the partner proteins (SecY(Figure 1D and Figure 1—figure supplement 4A) and PpiD (Figure 1F)). We believe that these results clearly validated our assignment of the XLs.

To show the correctness of the assignment of the XLs in Figure 2, we have added a new Figure as Figure 2—figure supplement 1B, in which the whole gel images of all the pulse-chase experiments shown in the new Figure 2 are presented. The data show that the XLs were generated in a UV-irradiation dependent manner, further confirming that they are pBPA-mediated XLs.

An example (a representative result) of the actual quantification procedure for the immunoprecipitated (IP) bands is shown in the new Figure 2—figure supplement 2 to help understanding by readers. All the raw data used for the quantification are presented in the Source data files. Regarding the Figure 2, all the quantified regions containing IP bands are shown by squares in the representative gel image on the second page of the file. All the quantified and static data of the experimental results have been summarized in the Excel files.

3) In addition, proof that SecA is involved as their model suggests would complete the story. This should also be a straightforward experiment considering the expertise of the authors.

As pointed out by the reviewer 2, we previously showed that the NaN_3_ (a SecA inhibitor) treatment of cells severely compromised both translocation and arrest-cancelation of VemP (Mori et al., (2018)). Although it strongly suggests involvement of SecA in the translocation of VemP, we understand that we have not demonstrated direct interaction between VemP and SecA, nor provided genetic evidence about involvement of SecA in the VemP translocation. According to the reviewer 2’s comment, we examined export of VemP in a secA51(Ts) mutant cell by pulse-chase experiments. Our results clearly showed that the translocation and arrest cancelation of VemP was severely and specifically retarded at high temperature in the secA51(Ts) mutant cells, but not in its isogenic wild type cells, indicating that SecA is crucial for VemP biogenesis and its arrest cancelation. We have added the new data as Figure 2—figure supplement 3 and inserted description of these results in the Results.

About the VemP-SecA interaction, several unidentified high molecular weight bands that might represent the VemP-SecA XLs have been observed in Figure 1—figure supplement 1. We examined these bands and found that an XL of ~150 kDa, observed with the VemP(L40pBPA) variant, was precipitated with an anti-SecA antibody. However, the PiXie analysis showed that the pulse-labeled SecA-VemP XL remained stable during the chase periods, in sharp contrast to XLs with other proteins shown in the Figure 2. These results suggest that, while the observed SecA-VemP cross-linking could show the ability of SecA to directly recognize VemP, it may not reflect their physiological interaction that occurs during the translocation of VemP. It would be possible that the off-pathway interaction between VemP and SecA can be stable and more easily detected than the physiological one that could occur in a very short time scale. Considering the above points, we did not analyze the XL of ~150 kDa any further. We have added these results as new Figure 1—figure supplement 2 and explained them in the Results in the revised manuscript.

Reviewer 1:1) All the VemP-interactors has been identified from the anticipation based on the molecular weights of the crosslinked products. Since the crosslinked products are so many other than those the authors selected, there would be other interactors that might be involved in the arrest/cancelation process. Please discuss.

Thank you for the comment. We basically agree with the reviewer's opinion. We understand that a number of extra bands (possible XLs) were detected with many VemP(pBPA) derivatives. It would be possible that some of the uncharacterized bands might represent the XLs with un-identified partner proteins that contribute to the arrest/cancelation process of VemP. We have added description on this point in the Results in the revised manuscript.

2) Is the proposed mechanism to pull the VemP nascent chain applicable to that for SecM?

As described in the manuscript, the arrest cancelation of VemP occurs at a late step in its translocation after the processing of its signal sequence, as a result of co-operative functioning of SecDF and PpiD. On the other hand, the arrest cancelation of SecM seems to occur at an earlier step of translocation, that is, before the processing of the signal sequence. At this step, SecDF and PpiD would not be able to participate in the translocation/arrest-cancelation of SecM. The regulation system for the SecM arrest cancelation would be suitable for directly monitoring function of SecA ATPase whose expression is controlled by SecM. In addition, SecM seems not to possess a residue corresponding to Arg-85 of VemP that plays a crucial role in the cancelation. Therefore, we have stated that the mechanism proposed in this paper would be utilized for the arrest-cancelation of VemP but not for that of SecM. We have briefly discussed the difference between VemP and SecM in theDiscussion in the revised manuscript.

3) The deletion of the ppiD gene causes no detectable defect of outer membrane biogenesis (Justice et al., 2005). Is the contribution of PpiD on the translocation specific to VemP?

Obviously PpiD is not essential for cell viability and biogenesis of major OMPs, as shown by Justice et al. However, their results would not contradict possible contribution of PpiD to transport of some minor secretory proteins. Since PpiD is well-conserved among enterobacteria including *E. coli* that possess no VemP protein, we think it possible that PpiD plays physiological/regulatory roles in biogenesis of some kinds of secretory proteins even in *E. coli,* like in the case of VemP. This point has been discussed in the Discussion in the revised manuscript.

4) Some of the labels in Figures (e.g., Figure 1—figure supplement 4, gel for secY-his10 strain) are incorrect.

We apologize for the mistakes. We have corrected the errors.

Reviewer 2:1) The authors find that Ffh, SecD/F, and PpiD all impact stall cancellation. The specificity of these observations would be greater if the authors could show that a translocon component that is not involved in VemP translocation does not impact the stall or cancellation. For example, does acute depletion of YidC have any effect? This experiment, if feasible, is important to exclude the possibility that any perturbation to protein translocation would impact VemP.

We completely agree with the above comment. It would be important to obtain data showing that a factor not involving in the VemP translocation does not contribute to the arrest cancelation of VemP. Although the reviewer requested us to examine the effect of YidC depletion on the VemP arrest cancelation, we have not performed this experiment as we think that it would be difficult to obtain a clear conclusion by this approach, from the following reasons. (i) A long-term cultivation (about 5-6 hours) will be required for effective depletion of YidC. Because YidC depletion will affect the biogenesis of many membrane proteins including the translocon components, we might observe side effects caused by a reduction of translocon-components after such longer depletion, even if YidC itself is not directly involved in VemP translocation. (ii) On the other hand, even when we observe no effect on the VemP translocation/arrest cancelation upon YidC-depletion, we cannot exclude the possibility that a small amount of YidC is sufficient for the above processes. (iii) indeed, though we have not detected a YidC-VemP XL so far, we cannot exclude the possibility that YidC contributes to VemP translocation.

Instead, we examined the effect of a deletion of the secB gene that encodes a secretion specific chaperone on the VemP arrest cancelation. It is well known that SecB is a member of the Sec translocation pathway and contributes to the post-translational target of substrates to the SecYEG translocon. Our data clearly show that the knock-out of the secB gene severely retards processing of the signal sequence of MBP, a SecB-dependent substrate, but not impact the VemP translocation and its arrest-cancelation. These results not only indicate that perturbation of some aspects of protein translocation does not necessarily affect the VemP translcation/arrest-cancelation, but also reinforce our conclusion that VemP is co-translationally targeted to the translocon via the SRP pathway. We have added these data as the new Figure 3—figure supplement 1 and its description in the Results in the revised manuscript. We thank the reviewer again for the productive suggestion.

2) The crosslinking result in Figure 4E showing the SecD-PpiD interaction would be more convincing and specific if the BpA were placed in another position in the periplasmic side of SecD/F and shown to NOT crosslink to PpiD. This would help exclude the possibility of promiscuous crosslinking from any periplasmic region of SecD/F due to its general proximity to the periplasmic chaperone PpiD.

Thank you for pointing out an important point. We have performed systematic in vivo pBPA cross-linking experiments targeted to many residues locating at the molecular surface of the first periplasmic domain of SecD, and found that only a specific set of the residues (Asp-359 presented in this paper and its neighboring residues) were cross-linked with PpiD, suggesting that a specific region in SecD is involved in its interaction with PpiD.

Since we want to publish the above cross-linking results elsewhere as a part of a separate paper, we would not like to include the results in this manuscript, if possible. However, if you need to see the data in order to review this work, we will be happy to disclose the unpublished data as a confidential information. If so, please let me know.

3) The authors propose that SecA is involved in translocation based on the effect of N3. However, they do not document an interaction of substrate with SecA or a genetic requirement. A substrate interaction would seem to be a strong prediction of their model. As a number of unidentified high molecular weight crosslinking products are observed in Figure 1—figure supplement 1A, it seems plausible that a direct interaction with SecA is indeed detectable. Have the authors checked this? If so, this would strengthen their study because all physical interactions show in their model would have experimental support (and it would really demonstrate how their PiXiE method can be comprehensive).

Thank you for the valuable comments. Please see our response to the reviewing editor’s comment 3.

4) The discovery of a specific role for R85 is the most intriguing finding in the study, but is hardly explored at all. While many experiments could be proposed that are beyond the scope of this study, the one experiment that I feel is within the scope is to document the (potential) relationship between R85 and translocation-dependence on SecD/F/PpiD. If I have understood the author's model, R85 partially 'clogs' in the SecYEG translocon, which makes translocation dependent on SecD/F/PpiD. It is this dependence on SecD/F/PpiD that makes VemP a monitor of a late step in translocation. It seems important to test whether R85 makes VemP translocation dependent on PpiD, independent of arrest. In this experiment, one could analyse W143A versus W143A/R85A for translocation in WT versus ∆PpiD strains. The aim here is to test whether R85 imposes PpiD-dependence to translocation (by monitoring signal cleavage). If so, this would indicate that VemP has evolved and retained R85 in order to make its translocation and arrest cancellation dependent on PpiD, as speculated by the authors.

Thank you for the productive proposal. Please see our response to the reviewing editor’s comment 1.

5) The colour schemes in Figure 1A and Figure 1—figure supplement 1B are not concordant (for example, uL22 is green in Figure 1A, but in Figure 1—figure supplement 1B green is a 100 kD crosslink; same for other partners). It would be more helpful to a reader if these colours matched up. Perhaps a good solution is to make a hybrid figure that combines all of these observations into one nice panel that is provided in the main figure as a revised Figure 1A. This way, a reader can appreciate an overview of all the crosslinks observed (i.e., Figure 1—figure supplement 1B) together with the subset that were verified by IPs (i.e., Figure 1A). One way to do this is to simply take Figure 1—figure supplement 1B, and under the respective sized crosslinks, put the corresponding protein and dots where this was verified by IPs (i.e., the information Figure 1A).

Thank you for your suggestions to improve our presentation. According to the suggestion, the colors of XLs in the Figure 1—figure supplement 1 was coordinated with the corresponding colors used in Figure 1A to increase clarity. In addition, we have modified the Figure 1 by adding a dashed red line that represents positions at which possible cross-linking with PpiD was observed in the Figure 1—figure supplement 1.

Although we deeply appreciate your helpful suggestion, we would like to show the data in the Figure 1A and those in the Figure 1—figure supplement 1B as separate figures, after introduction of several modifications described above. It is because some XLs observed in the Figure 1—figure supplement 1 would represent non-physiological interactions, as we explained above for the SecA-VemP XLs. We think that the separate presentation of the overall crosslinking profile as revealed by anti-VemP IP (Figure 1—figure supplement 1B) and the XLs that would reflect physiological interaction with the conformed partners (Figure 1A) would facilitate correct understanding of the crosslinking data by readers.

6) In the Discussion, the authors claim that their uL22 crosslink is in vivo validation of specific interactions seen by cryo-EM. This conclusion is probably overstated. The photo-cross-linking documents close proximity, but cannot be used to infer specific residue contacts. Perhaps reword this to say that the crosslinking "are consistent with" the previous structure.

Thank you for the suggestion. It is true that the crosslinking does not necessarily indicate the direct physical contact between the crosslinked proteins. We have changed the text in the Discussion in the revised manuscript, as suggested.

Reviewer 3:1) Most of the experiments were performed by expressing V. alginolyticus VemP in E. coli as heterologous host. I assume that this is necessary because an orthogonal tRNA/tRNA synthetase system for pBpa insertion does not exist for V. alginolyticus. This should be mentioned. It could also help if the authors would briefly summarize what is known about SecYEG, SecA, SRP etc in V. alginolyticus. This could strengthen their arguments that the data they observe in *E. coli* indeed reflect what likely is happening in the native host.

Thank you for the valuable comments. According to the suggestions, we have made several additions and modifications to the text in the last part of Introduction, which includes explanation of our heterologous experimental system, brief introduction of the Sec system in *Vibrio* species and a statement about significance of this study even using the heterologous system.

2) Ffh was found to cross-link to only one position within the signal sequence (A15), which I found surprising (Figure 1E), considering that approx. 15 residues of the signal sequence were tested (Figure 1—figure supplement 1). Residue A15 and H22 also show cross-linking products of 150 kDa and > 150 kDa, what are those? The predicted Ffh cross-link is also very weak and not detected by the VemP antibodies. Only after pull down with Ni-NTA reacting with His-tagged Ffh, the cross-link is visible. The predicted Ffh-VemP cross-link migrates at 55-60 kDa, which does not seem to fit the expected size (50 kDa + 15-17 kDa)? Are the authors really sure that this reflects an SRP cross-link?

With respect to the VemP-Ffh crosslinking, we quantitated the signal intensities of the regions corresponding to the migrating position of the VemP(A15pBPA)-Ffh XL for all the lanes in the gel shown in Figure 1—figure supplement 4B and found that several VemP derivatives containing pBPA at the positions 11, 18, 31, 36 and 46 gave slight but significant signals, suggesting that crosslinking with Ffh occurs not only at the A15 position but also at the other positions within or adjacent to the signal sequence. Thus, we suppose that the Ffh can interact with the VemP signal sequence at several positions (although the interaction at the position A15 might play a major role in this interaction).

It is sometimes observed that apparent molecular sizes of XLs estimated from their mobilities on SDS-PAGE don’t well match their sizes calculated as the sum of molecular sizes of the cross-linked proteins. We think that mobilities of XLs on SDS-PAGE depend on not only molecular sizes but also their shapes (ex. a T-like shape or a X-like shape). Probably, the Ffh-VemP XL would exhibit an anomalous mobility due to the above reason. We concluded the XL observed for VemP (A15pBPA) to be the XL with SRP(Ffh), from the following reasons. (i) The XL was reproducibly and specifically pulled down by Ni-NTA agarose, only when we used the cells expressing His-tagged version of Ffh. (ii) The XL was significantly enhanced by compromising the targeting step of VemP using NaN3 treatment of the cells (See Figure 2—figure supplement 4), which is expected to stabilize the VemP-SRP interaction. (iii) The generation of the VemP-SRP XL is consistent with our genetic/biochemical data that SRP contributes to both VemP translocation and its arrest-cancelation.

3) It would have been expected that the authors also observe XL to SecA, which based on their model should interact with the mature domain of VemP. Has this been tested and have the high MW XL at 150 kDa and > 150 kDa (Figure 1—figure supplement 1) been probed with anti-secA antibodies?

Thank you for the comments. The reviewing editor and the reviewer 2 also raised similar questions. Please see our response to the reviewer editor’s comment 3.

4) Considering their model, it would have been expected that SecY cross-links also to the signal sequence, if this is trapped at the lateral gate during translocation. Trapping of the signal sequence and a hairpin-like insertion of the nascent VemP would explain the cross-links to PpiD at residues 30-60 and cross-links to SecY of down-stream residues. This could suggest that these down-stream residues are inside the SecY channel. Then why do we see SecG? How does this fit to the general geometry of the SecYEG translocon?

As discussed above, the signal sequence of VemP should interact with the lateral gate of SecY at an early step of its translocation and could generate a XL, when pBPA is introduced into the VemP signal sequence. However, the signal sequence of VemP is mostly removed at an early time point of translocation, where translation of VemP is still arrested (note that even after pulse-labeling, the arrested VemP molecules are mostly in the processed form). Thus, even if the pBPA-containing signal sequence of VemP, which had been cleaved off form AP-un, is crosslinked with the SecY, the XLs cannot be isolated by immunoprecipitation using the anti-VemP antibody raised against the epitope in the mature part of VemP. Although the XLs might be able to be isolated by using a His-tag attached to SecY, a small size of the signal sequence will cause only a slight retardation in the mobility of the XL bands, making it difficult to distinguish them from a broad SecY band. Also, a low amount of AP-un having pBPA in the signal sequence will hinder detection of its crosslinking with SecY.

Concerning the cross-linking with SecG, the crystal structure of the SecYEG complex determined by Tanaka et al., (2015) reveals that the transmembrane region of SecG is located on the exterior surface of the N-terminal half of the SecY channel, and the cytoplasmic loop between the TMs 1 and 2 of SecG covers the cytoplasmic cavity of the SecY channel at a resting state. It would be thus conceivable that even at the active state, the cytoplasmic loop of SecG exists near the cytoplasmic cavity of SecY. This would provide one explanation for why SecG as well as SecY is cross-linked with a translocating VemP.

Although we appreciate your comments, which should be useful and very interesting to the researchers working in this field of study, we are afraid that inclusion of the discussion on this point in the main text may interrupt the central story of this study and thus hinder its understanding by many of readers of this journal that would be mostly non-specialists. Thus, we would like to include the above discussions as the Appendix 1, if possible.

5) The sequential interaction profile of VemP is very interesting, although I think one has to be very careful with the interpretation: there is a substantial variation in the amount of AP-un, AP-pro and FL VemP for the different pBpa variants displayed in Figure 2. Therefore, the rather small differences in the decrease of XL products could simply reflect these variations.Thus, although their interpretation is probably correct and fits to the general concept of co-translational targeting, the quantification shown in Figure 2 likely overestimates the accuracy of the cross-linking method.

We appreciate your comments. We agree that we cannot completely exclude the possibility that difference in the rates of disappearance of these XLs is not due to difference in the timing of the interactions, but is due to variations in the amounts of the VemP(*p*BPA) derivatives. However, the amounts of AP-un, AP-pro and FL VemP are comparable among the VemP(*p*BPA) derivatives used in the kinetic study. Also, while we analyzed only one XL for each of uL22 and Ffh as we unfortunately detected a clear crosslinking at a single position for these proteins, two different XLs were used for kinetic analysis of each of SecG and PpiD, and they gave consistent results. In addition, as pointed out by the reviewer, our results are nicely fit with the general concept of co-translational targeting. We thus think that our interpretation is well grounded, but modified the relevant text to make it a more careful description. Specifically, we have changed the words “We demonstrated” in the original manuscript to “The kinetic analyses suggest” in the Discussion in the revised manuscript.

6) I understand that the authors found no XL to YidC, although it is in close contact to SecY and has like PpiD a large periplasmic domain. The authors might want to refer to the papers of Götzke et al., (2014) and Jauss et al., (2019), who suggested that PpiD and YidC associate with distinct SecYEG populations. This could explain why PpiD but not YidC is found in their cross-linking approach.

Thank you for the information. It could be a reason why we did not detect a VemP-YidC XL. We have discussed this point in the Appendix 1 described above.

7) The authors should explain why PpiD shows two cross-linking products. Does this reflect XL to AP-un and AP-pro? The same applies to Figure 4E: why does SecD359 shows two PpiD cross-linking products of significant size difference (approx. 25 kDa)?

We think that *p*BPA introduced at position 359 in SecD can be cross-linked with two different positions in PpiD that could be located distantly in the primary structure. As described above, it is often observed that XLs generated from the same two proteins exhibit different gel mobilities when their crosslinking sites are different. This is briefly explained in the Results.

8) The use of the secY24/pSTV28-syd strain as translocation defective strain is probably not the best choice. Why not using a more defined SecY mutant, like the SecY39 or SecY40 strains that have been characterized in Koreaki Itos lab and other labs. The partial suppression of the secY24 mutant by overexpression of the largely uncharacterized protein Syd introduces an ambiguity that could be easily prevented. Along the same line: if the secY25/pSTV28-syd is indeed translocation deficient, isn't it surprising that full-length VemP and processed VemP/MBP are detectable at all?

Thank you for the critical comments. We have previously demonstrated that the *secY24/psyd* system used in this study as well as the *secY39* mutation severely compromised translocation of VemP and stabilized its arrested form (Mori et al., (2018)). We indeed observed small amount of full-length VemP and processed VemP/MBP in the *secY24*/p*syd* system. However, similar results were obtained, even when the *secY39* mutation was used (Please compare of Figure 1B, lanes 1 to 3 and 10 to 12 (*secY24/psyd*) with Figure S2A (*secY39*) in the above Mori et al., paper), indicating that complete inhibition of protein translocation is quite difficult using these systems (actually the *secY24/psyd* system appears to exert slightly stronger effects than *secY39*). As you mentioned, inactivation of the Sec system using the *secY24/psyd* was not complete, but we think it sufficient to evaluate the effects of the mutations of VemP on its translocation. Indeed, we were able to detect clear difference between the behaviors of VemP(W143A) and VemP(R85W) under the SecY24/Syd-induced translocation-compromised condition. These previous results and observations, we think, justifies the utilization of the *secY24/psyd* system.

9) A general comment regarding the figures and XLs: the color of the arrow heads is difficult to see (with the exception of red) and the color code in the main text and Figure 1—figure supplement 1B does not match.

We have changed the color of the arrow heads. Details have been described in the response to the reviewer 2’s comment 5. Please see above.

In addition, please note that we have corrected both the average values of the Ara (-) sample (shown as purple squares) and the lengths of the corresponding error bars in the graph of the Figure 3A, because we found mistakes in the quantified data used for preparation of the original Figure 3A. The corrections do not affect the conclusion. We apologize for the careless mistakes.

We have also included the source data for Figure 2, Figure 3 and Figure 4 and Figure 2—figure supplement 1, Figure 2—figure supplement 3 and Figure 2—figure supplement 4 and Figure 5—figure supplement 1 and Figure 5—figure supplement 2.